



# Contribution of marine biological emissions to gaseous methylamines in the atmosphere: an emission inventory based on satellite data

Qi Zhang[2], Shiguo Jia[1, 4], Weihua, Chen[3], Jingying Mao[8], Liming Yang[5], Padmaja Krishnan[6], Sayantan Sarkar[7], Min Shao[3], Xuemei Wang[3]

[1] School of Atmospheric Sciences, Sun Yat-sen University and Southern Marine Science and Engineering Guangdong Laboratory (Zhuhai), Zhuhai, 519082, China
[2] Tianjin Air Pollution Control Laboratory, Tianjin Academy of Eco-Environmental Sciences, Tianjin, 300191, PR China
[3] Guangdong-Hongkong-Macau Joint Laboratory of Collaborative Innovation for Environmental Quality, Institute for Environmental and Climate Research, Jinan University, Guangzhou, 510632, China
[4] Guangdong Provincial Field Observation and Research Station for Climate Environment and Air Quality Change in the Pearl River Estuary, Guangzhou, 510275, China
[5] Department of Chemical and Biomolecular Engineering, National University of Singapore, Singapore, 117576, Singapore
[6] Division of Engineering, New York University Abu Dhabi, Abu Dhabi, P.O. Box 129188, United Arab Emirates
[7] School of Engineering, Indian Institute of Technology (IIT) Mandi, Kamand, Himachal Pradesh 175075, India
[8] Scientific Research Academy of Guangxi Environmental Protection, Nanning, 530022, China
*Correspondence to*: Xuemei Wang (eciwxm@jnu.edu.cn), Shiguo Jia (jiashg3@mail.sysu.edu.cn)

**Abstract.** Methylamines can readily react with acidic gases in the atmosphere, which consequently has an important impact on the atmospheric environment. It is difficult to measure amines in field studies due to their high reactivity, and therefore, numerical modelling is an effective tool to study ambient amines. However, the contribution of marine biological emissions (MBE), an important source of methylamines (MA), has been insufficiently investigated in the current emission inventory. Therefore, this study utilized satellite data such as Sea Surface Temperature (SST), Chlorophyll-a (Chla), Sea Surface Salinity (SSS) and model simulation data (Wind Speed, WS) to establish a more reasonable MBE inventory of amines. Spatial and temporal distribution of methylamine emissions indicates that MBE fluxes of monomethylamine (MMA) and trimethylamines (TMA) can be comparable with or even higher than that of terrestrial anthropogenic emissions (AE), while for dimethylamines (DMA), the ocean acts as a sink. The method used in this study can better reflect the exchange direction of amines between ocean and atmosphere, and reflect the emission characteristics of different amines. From WRF-Chem simulation results, the concentration of amines in the coastline was found to increase significantly due to the contribution of MBE. Wind and Chla were potentially the most important factors affecting MBE fluxes. WS is directly used in the calculation of ocean-atmosphere exchange coefficient $K_g$, and the direction of the prevailing winds in different seasons affects the area of influence of the MBE. Chla indirectly influences the calculation results of exchange flux by affecting the calculation of pH. In addition, the emission fluxes and spatial distribution of AE and wet deposition also affect the simulation of amines.



## 1 Introduction

Amines are derivatives of ammonia, wherein the hydrogen atoms are replaced by hydrocarbon groups. The presence of alkyl, acyl and other functional groups in the amines induces dipole moment effects and makes amines more alkaline than ammonia. Hence, similar to ammonia, amines can readily react with acidic gases in the atmosphere (e.g., gaseous sulfuric acid and organic acids) (Barsanti et al., 2009; Hong et al., 2018; Miao et al., 2018), and can also be substituted for $NH_4^+$ undergoing gas-to-particle phase conversion (Qiu et al., 2011; Qiu and Zhang, 2013; Waller et al., 2018). Amines are known to be important precursors in the nucleation stage during the formation of new aerosol particles in both laboratory experiments and field observations. For example, Elm et al. (2016, 2017) proved that acid–base nucleation requires strong alkaline precursors, with dimethylamines (DMA) and diamines assuming predominance compared to ammonia. Wang et al. (2018) investigated the synergistic effect of methylamine and ammonia during the nucleation process, converting the existing nucleation models from "ternary nucleation" to "quaternary nucleation". Additionally, Yao et al. (2018) indicated based on field observation and CLOUD (Cosmic Leaving Outdoor Droplets) chamber simulations that the "$H_2SO_4$–DMA–$H_2O$" ternary nucleation system may be the primary nucleation mechanism in typical urban areas like Shanghai. Interestingly, Liu et al. (2018) found a strong linear relationship between total amines and $NO_3^-$ in $PM_{2.5}$, implying that amines mainly exist in the form of aminium nitrate. Lian et al. (2020) recently reported that due to differences in their physicochemical properties, diethylamine (DEA) and trimethylamines (TMA) exhibit variable distributions over different particle types and distinct mixing states; for example, DEA was found to be more associated with sulfate compared to TMA.

Generally, amines can be emitted from sources such as animal husbandry, industry, combustion, composting operation, automobiles and other anthropogenic emissions (AE). They are also emitted from natural sources including the ocean, biomass burning, vegetation and geologic processes (Ge et al., 2011). There are about 150 known amines in the atmosphere, including aliphatic amines, amides, aromatic amines, piperazine, pyrrolidine, piperidine, pyrrole, pyridine, pyrazine, quinoline, nicotine, indole, skatole and related derivative amides (Smith et al., 2008, 2010; Loukonen et al., 2010; Barsanti et al., 2009; Ge et al., 2011; Ma et al., 2019; Elm et al., 2016, 2017). Among the above amines, low molecular weight methylamines (popularly referred to as aliphatic amines) containing carbon number in the range of 1–6 are the most abundant species (Ge et al., 2011; Zheng et al., 2015; Yao et al., 2016). Data collected from field observations in urban and mountain sites in southern China showed that monomethylamine (MMA) and DMA were the dominant amines, together contributing to approximately 70% of gaseous amines and 80% of particulate amines (Liu et al., 2018).

Globally, ammonia and amines emissions are about 50000±30000 and 285±78 Gg N $a^{-1}$, respectively (Ge et al., 2011). It is believed that amines have stronger reactivity and are more easily oxidized by oxidants ($\cdot NO_3$, $\cdot OH$ and $O_3$), condensed into particulates and scavenged from the atmosphere by both wet and dry depositions (Carl and Crowley, 1998; Barsanti et al., 2009; Qiu et al., 2011; Qiu and Zhang, 2013; Yu and Luo, 2014; Yao et al., 2016; Mao et al., 2018; Waller et al., 2018). As a result, the concentration of amines in the atmosphere is much lower (usually 3 order of magnitude) than that of ammonia and thus more difficult to be measured (Ge et al., 2011; Yu and Luo, 2014; Yao et al., 2016). For example, the total





concentration of MMA, DMA and TMA was 7.4±4.7 pptv, much lower than the concentration of ammonia (1.7±2.3 ppbv),
in suburban areas of Nanjing measured during the summer of 2012 (Zheng et al., 2015). The concentrations of $C_1$–$C_6$ amines
measured by Yao et al. (2016) in the urban areas of Shanghai in the summer of 2015 were 15.7±5.9, 40.0±14.3, 1.1±0.6,
15.4±7.9, 3.4±3.7, and 3.5±2.2 pptv, respectively.
Typical urban areas along China's east coast have a complex background atmospheric pollution, and in addition to the
complex AE from the mainland, the contribution of ocean sources cannot be ignored. Zhou et al. (2019) detected relatively
high concentrations of dimethylaminium ($DMAH^+$) and trimethylaminium +diethylaminium ($TMDEAH^+$) in aerosol samples
from Huaniao Island, an ocean site near the Yangtze River Delta, and estimated that marine biogenic emissions (MBE)
contributed to 26–31% and 53–78% of aerosol aminiums over Huaniao Island in the autumn of 2016 and summer of 2017,
respectively. Xie et al. (2018) carried out three inland campaigns and one sea-beach campaign in Qingdao, along with five
marine campaigns in marginal seas of China and the northwest Pacific Ocean. They observed increased concentrations of
$DMA^+$ and $TMA^+$ in $PM_{0.056-10}$, particularly, the ratios of $DMA^+/NH_4^+$ and $TMA^+/NH_4^+$ increased by 1–2 orders of
magnitude in the marine and sea-beach atmospheres, indicating that the overwhelming majority of amines were derived from
marine sources. Further, Hu et al. (2018) linked the significant increase of primary TMA+ and secondary DMA+ in particles
to the emissions of sea-spray aerosols and gaseous precursors from various cyclonic eddies.
The atmospheric background in eastern China is generally composed of high concentrations of pollutants and strong oxidants,
leading to the formation of secondary pollutants which could be the main reason for the poor air quality (Huang et al., 2014).
The secondary gaseous acids, in turn, react with alkaline gases (e.g., $NH_3$, amines, etc.) and water molecules ($H_2O$), enabling
fast gas-to-particle nucleation, and forming a large number of molecular clusters (Wilemski et al., 1983; Kulmala, 2003;
Kirby et al., 2011; Qiu et al., 2011; Qiu and Zhang, 2013; Waller et al., 2018). The rise in aerosol number and mass
concentration caused by the formation and subsequent growth of new aerosol particles acts as an impetus for the occurrence
of serious haze pollution events in China (Gao et al., 2011; Wang et al., 2014; Xiao et al., 2015; Zhang et al., 2015; Wang et
al., 2016; Wang et al., 2017; Wu et al., 2017; Yu, 2011). In addition to the contribution from AE, eastern China's
geographical location on the east coast of Eurasia makes it impossible to ignore the impact of MBE. Due to its unique
geographical location and pollution composition, eastern China was chosen as the research area in this study.
Emission composition and spatiotemporal factors affect the concentration of gaseous amines and the formation mechanism
of new particles in polluted areas such as in China. Numerical modelling is an effective tool to study the gaseous amines in
the atmosphere. And accurate emission inventories of amines can lead to better model simulations of its atmospheric levels
and corresponding spatiotemporal variations, and it in turn can lead to a better understanding of amines' role in atmospheric
chemistry and aerosol formation. However, few studies have employed regional and global scale models to simulate gaseous
amines and mostly adopted the fixed emission mass ratio of amines to ammonia in the emission inventory (Myriokefalitakis
et al., 2010; Yu and Luo, 2014; Bergman et al., 2015). The assumption that the ratio of amines to ammonia emissions is
fixed for all emissions is obviously too simplified. Therefore, Mao et al. (2018) derived source-dependent ratios (SDR) that
distinguished $C_1$-amine ($CH_3NH_2$, also known as MMA), $C_2$-amine ($C_2H_7N$, also known as DMA) and $C_3$-amine ($C_3H_9N$,



also known as TMA) emissions from five different source types (agriculture, residential, transportation, chemical industry, and other industry), which is the most reasonable emission inventory of amines to date. However, the emissions mass ratio obtained was still arbitrary thus leading to high uncertainty in the simulation results. To overcome the arbitrariness of the traditional amine emission inventories, this study establishes an emission inventory of amines including MMA, DMA, and TMA from MBE based on their mechanisms of production. Model simulated wind speed and Multiple satellite datasets including chlorophyll-a, ocean surface temperature and salinity were incorporated to calculate the emission fluxes of amines.

## 2 Data and Methods

### 2.1 Study area

Our study area, shown in Fig. 1, was selected to effectively quantify the impact of MBE in the world's largest ocean, the North Pacific, on China. The selected region encloses longitudes from 85.2° E to 140.8° E and latitudes from 15.4° N to 51.9° N. The land area includes most of China, as well as the Korean peninsula and the southern part of the Japanese archipelago. In addition to the Bohai Sea, the Yellow Sea, the East China Sea and the South China Sea, the maritime area also includes the Sea of Japan and parts of the Pacific Ocean.

### 2.2 Anthropogenic Emission Inventory

Terrain AE inventories are adopted from the Multi-resolution Emission Inventory for China (MEIC) (http://www.meicmodel.org/), which was developed by Tsinghua University, and spatially allocated into the 0.25° × 0.25° grid cells. Types of emissions include transportation emissions, residential emissions, other industrial emissions, power plants and agricultural emissions. The primary pollutants mainly include sulfur dioxide ($SO_2$), nitrogen oxides ($NO_x$), carbon monoxide (CO), ammonia ($NH_3$), particulate matter of size less than 10 μm ($PM_{10}$), particulate matter of size less than 2.5 μm ($PM_{2.5}$), black carbon, organic carbon, and VOCs. To establish the MMA, DMA and TMA emission inventories, we referred to the amines/$NH_3$ mass emission ratio for different types of emission sources from Mao et al. (2018), which are shown in Table 1. Compared with the previous method where a fixed ratio was used for the total emission sources, the method established by Mao et al. (2018) which provides different ratios for different emission sources has considerable advantages (e.g., it substantially improves the performance of the model in capturing the observed concentrations of MA). However, Mao et al. (2018) also believed that there was underestimation of the ratios they established. Since this study is only a preliminary attempt to explore the contribution of MBE to ambient amines, the method proposed by Mao et al. (2018) is adopted to establish an AE inventory of land-based amines emissions.

The emission inventory from marine ships is taken from the Tsinghua University database, with a horizontal resolution of 0.25°× 0.25° (Xiao et al., 2018) and pollutants mainly include $SO_2$, $NO_x$, CO, $NH_3$, $PM_{10}$, $PM_{2.5}$, black carbon, organic carbon, and VOCs.



### 2.3 Model setup and configurations

The online model, WRF-Chem, enables complete online coupling of the chemical module and the atmospheric chemical transport module to simulate trace gas and aerosol emissions, spatial transport, meteorological changes and chemical reactions. WRF-Chem has been widely used in the simulation of air pollutants, and its performance has been widely recognized (Cui et al., 2015; Chen, et al., 2019; Liu, et al., 2017). In this study, WRF-Chem version 3.7.1 was used to simulate the contributions of MBE and AE to amines. Physical and chemical parameterization schemes are shown in Table 2. It should be noted that ambient amines are mainly removed from the atmosphere through the following three ways: i) dry and wet deposition, ii) gas-phase oxidation with oxidizing substances (e.g., $\cdot OH$, $O_3$ and $NO_x$), and iii) aerosol uptake (Yu and Luo, 2014; Bergman et al., 2015). In this study, only the effects of gas-phase reaction and wet deposition on ambient amines concentration were considered in WRF-Chem. As per Yu and Luo (2014), the main oxidant for amines in the atmosphere is $\cdot OH$, and the effects of $O_3$ and $NO_x$ are not significant. Therefore, the reaction between amines and $\cdot OH$ alone was included in the CBMZ gas-phase chemical scheme adopted in this study, and the hydroxyl radical reaction rate constants ($K_{OH}$, $cm^3$ $mol^{-1}$ $s^{-1}$) were $2.2 \times 10^{-11}$, $6.5 \times 10^{-11}$, and $6.1 \times 10^{-11}$ for MMA, DMA, and TMA, respectively. The wet deposition of amines was treated in the same way as $NH_3$.

The simulation area is shown in Fig. 1. In the detailed configuration of the model, the centre was set at 28.5° N, 114.0° E, while the projection was set to Lambert. The year 2015 was chosen as the base year for simulations since it had more consecutive days with field observation (e.g., Zheng et al., 2015; Yao et al., 2016, 2018, etc). January, April, July and October are the representative months of the four seasons in China. However, observation data for gaseous amines for the entire month is unavailable. In this study, observation data from Yao et al. (2018) collected in Shanghai were used for verification, but the data only had daily mean values and some data were missing. Therefore, to understand the seasonality of MBE contributions, the days of four representative months with more consecutive observation data were selected as the simulation period of this study. The observation periods for January (Jan, from 2015.1.1 00:00:00 to 2015.1.10 18:00:00), April (Apr, from 2015.4.8 00:00:00 to 2015.4.17 18:00:00), July (Jul, from 2015.7.22 00:00:00 to 2015.7.31 18:00:00), and October (Oct, from 2015.10.9 00:00:00 to 2015.10.17 18:00:00) with more consecutive data were selected corresponding to winter, spring, summer and autumn, respectively. In addition, the chemical boundary used in this study was modelled using CAM-CHEM as a global simulation result. However, since the chemical species of CAM-CHEM did not contain amines, the three amines involved in this study, MMA, DMA and TMA, were obtained based on their mass ratio with ammonia, which were $8.5 \times 10^{-4}$, $1.56 \times 10^{-3}$ and $3.7 \times 10^{-4}$, respectively (Zheng et al., 2015). More specific information is available at https://www.acom.ucar.edu/cam-chem/cam-chem.shtml. Biogenic emissions were calculated online using MEGAN (Model of Emissions of Gases and Aerosols from Nature) coupled in the WRF-Chem (Guenther et al., 2006).


**2.4 Calculation of air–ocean net exchange flux of methylamines**
Part of ammonia and methylamines dissolved in seawater exist as solvated cations due to ionization (e.g., $NH_4^+$, $MMAH^+$,
$DMAH^+$, $TMAH^+$), accounting for >90% of the total dissolved ammonia and amines in seawater (Carpenter, et al., 2012). In
addition, there are a small number of unionized gas molecules (e.g., $NH_3$, $CH_3NH_2$, $(CH_3)_2NH$, $(CH_3)_3N$) that can be
exchanged between the atmosphere and the ocean (Carpenter, et al., 2012), and the net fluxes of ammonia and amines across
the interface can be calculated from the following equation (Liss and Slater, 1974):
$F = K_g\left([C_{(g)}] - H[C_{(s)}]\right)$    (1)
Where, $F$ is the gaseous interfacial flux between air and ocean (pmol m$^{-2}$ s$^{-1}$); $C_{(g)}$ is the concentration of gas (nM; e.g.,
$NH_{3(g)}$, $MMA_{(g)}$, $DMA_{(g)}$, $TMA_{(g)}$ determined by the measurement carried by Gibb et al., (1999); Table 3); $C_{(s)}$ is the
corresponding concentration of unionized solute molecules (nM); $K_g$ is the air–ocean transfer velocity (m s$^{-1}$); $H$ is the
dimensionless Henry's Law constant (the ratio of the partial pressure of the gas to the amount of gas dissolved in a
quantitative liquid at a certain temperature at equilibrium). The Henry's Law constant, $H$, depends on the temperature ($T$, °C)
and the ionic strength ($I$, mol dm$^{-3}$) of seawater. For $NH_3$, $H$ is calculated from Eq. (2). In this study, $H$ of amines was also
calculated based on Eq. (2) (Gibb et al., 1999).
$H = \left[28.0 exp\left(lnK_{H(293)} - \frac{4092dT}{T^2}\right)\right]^{-1}$    (2)
$K_H$ (M atm$^{-1}$) represents the temperature dependence of $H$ for transfer between the gaseous and liquid phases (Gibb et al.,

176    1999).

$lnK_H = 4092/T - 9.70$    (3)
The calculation of $K_g$ is shown below (Duce et al., 1991),
$K_g = \frac{U}{770+45MW^{1/3}}$    (4)
Where, MW is the molecular weight of the gas (g mol$^{-1}$), and U is wind velocity (m s$^{-1}$).
Solute molecules of ammonia and amines generally have the following ionization equilibrium in seawater (e.g., ammonia),
$[NH_{3(s)}] + H_2O \Leftrightarrow [NH_4^+] + [OH^-]$
The corresponding base dissociation constant ($K_b$) is calculated as follows:
$K_b = \frac{[NH_{4(s)}^+][OH^-]}{[NH_{3(s)}]}$    (5)
Consequently, Eq. (6) can be used to calculate the concentration of $C_{(s)}$ (Van Neste et al., 1987),
$[C_{(s)}] = \frac{[C_{(s)tot}^+][OH^-]}{K_b+[OH^-]}$    (6)
Where $[C_{(s\ tot}^+] = [C_{(s)}] + [C_{(s)}^+]$, is the total dissolved concentration of analyte (nM, i.e. $NH_4^+{}_{(s)tot}$, $MMAH^+{}_{(s)tot}$, $DMAH^+{}_{(s)tot}$,
$TMAH^+{}_{(s)tot}$), $[C_{(s)}^+]$ is the concentration of corresponding solvated cation (nM, e.g. $NH_4^+{}_{(s)}$, $MMAH^+{}_{(s)}$, $DMAH^+{}_{(s)}$,
$TMAH^+{}_{(s)}$). Since the data of amine dissolution in surface waters near the eastern coastal areas of China are scarce, this study
selected the average of the observed values of other sea areas (Table 3). Nakano and Wakanabe's (2005) research showed the


spatio-temporal distribution diagram of marine environmental parameters (sea surface temperature (SST), Chlorophyll a
(Chla), and pH) in the North Pacific Region. The results showed that marine environment near the mainland and marine
parameters are affected by the climate zone they are located in as well as the human activities near the coast. The eastern
coastline of China spans multiple temperature zones and the economic activities and concentrated population near the coastal
areas result in higher pollutant emissions. Therefore, observations at a single site cannot represent the dissolution of amines
in surface seawater off the eastern coastal areas of China. The research data listed in Table 3 are respectively from the
Pacific Ocean, Indian Ocean and Atlantic Ocean, corresponding to temperate, subtropical and temperate zones, respectively,
and all sites are located in densely populated areas. Therefore, the average of these studies is assumed as the concentration of
total dissolved amines. Hydroxide ion concentration ($[OH^-]$) in Eq. (6) can be calculated based on the measured pH,
$$[OH^-] = 10^{-(14-pH)} \tag{7}$$
Since this study is mainly focused in the Pacific region, which has a significant impact on the southeast coast of China, we
adopted a simple function for pH calculation proposed by Nakano and Watanabe (2005), which is applicable to the surface
seawater over the North Pacific Basin for all seasons, and is related to SST (same as $T$, K) and $Chla$ (mg m$^{-3}$),
$$pH_{p(T)} = -0.00266 SST - 0.0243 Chla + 8.892 \tag{8}$$
The dissociation constant $K_b$ in Eq. (5) is calculated from the corresponding stability constant, $pK_a$,
$$K_b = 10^{-(14-pK_a)} \tag{9}$$
Further, $K_a$ is also dependent on the temperature and ionic strength. For the NH$_{3(s)}$–NH$_4^+{}_{(s)}$ system, the calculation of $pK_a$ can
be obtained from the empirical equation provided by Khoo et al. (1977) and applicable when the water salinity is within 45:
$$pK_a = pK_{w_a} + (0.1552 - 0.003142\,T)I \tag{10}$$
Where, $pK_{wa}$ is the value of $pK_a$ in the pure water (T=20 °C, I=0 mol dm$^{-3}$). The relationship of $pK_a$ with T and I (Eq. (11)),
is also assumed for the amines. Gibb et al. (1999) has confirmed the usability of this method in the Arabian Sea and the
detailed information of the physicochemical parameters used in the above calculations are summarized in Table 4. Ion
intensity ($I$) can be calculated from the measured sea surface salinity (SSS) using the following equation (Lyman and
Fleming, 1940),
$$I = 0.00147 + 0.01988\,SSS + 2.08357 \times 10^{-5} SSS^2 \tag{11}$$

### 216  2.5 Data Used in the Calculation

To calculate net exchange flux of methylamines between air and ocean, the key is to calculate the concentration of amines
dissolved in seawater. As a class of soluble organic bases, the dissolution and ionization equilibrium of MMA, DMA, and
TMA are closely related to the pH of seawater. Therefore, in this study, high-precision satellite data were used and the
algorithm suitable for the north Pacific region proposed by Nakano and Watanabe (2005) was employed to calculate the
high-precision spatial and temporal distribution of pH data of the studied marine region.





### 2.5.1 Sea Surface Temperature

SST is one of the most important variables in marine hydrological conditions. It can directly or indirectly affect MBE of amines by influencing biological and physicochemical processes in the ocean. The global high-resolution satellite-based daily mean SST data used in this study was released by NOAA's National Environmental Satellite Data and Services (NESDIS) office, with a resolution of 0.05° (~5km) and within operational bias. The SST data was combined with analysis from the US, Japanese and European Synchronous Infrared Imagers and Low Earth Orbit Infrared (US and Europe) to produce a set of high-resolution (5km) SST data products. The grid scale was chosen to achieve a resolution close to the Nyquist sampling standard of the Rossby radius (~20 km) at mid-latitudes to preserve mesoscale oceanic features such as vortices and frontal meanders. The input SST data itself is also internally processed using the Geo-SST Bayesian and physical retrieval methods (GOES-E/W, Meteosat-10). For polar orbits and Himawari-8, the Advanced Clear Sky Processor for Oceans (ACSPO) is adopted. More detailed information is available at https://coastwatch.noaa.gov/cw/satellite-data-products/sea-surface-temperature/sea-surface-temperature-near-real-time-geopolar-blended.html. The SST data has been updated since 2019. In this study, the average of the same month in the simulation period was selected for calculation. The specific spatial and temporal distribution is shown in Fig. 2(a, b, c, d). SST presents the seasonality of July > October > April > January. In terms of spatial distribution, the SST of Bohai Sea and Yellow Sea in China is lower than that of sea areas with the same latitude in spring and winter, which may be due to the fact that it is controlled by continental high in spring and winter, and the prevailing northwest dry and cold monsoon, leading to low temperature. Similarly, this can also explain the low SST in the coastal areas, and with the decrease of latitude, the influence of the continental high gradually decreases, which is similar to that in the sea areas of the same dimension in Guangdong, Hainan and other provinces.

### 2.5.2 Chlorophyll-a

The Ocean Color Satellite Sensor measures visible light at specific wavelengths that leave the ocean surface and reach the top of the atmosphere where the sensor is located. Measurements of these visible spectra, along with simultaneous measurements of near-infrared (NIR) and short-wave infrared (SWIR) wavelengths, can be used to calculate ocean color or standardized radiation out of water (nLw). In turn, nLws can be used to derive other marine properties, such as the concentration of Chla (chlorophyll is mainly used for photosynthesis, so it can be used as an important indicator of phytoplankton biomass in seawater) and the extinction coefficient of infiltration irradiation (Kd(PAR) and Kd(490), which are related to the purity of seawater). The Chla datasets used in this study were obtained from the Visible and Infrared Imaging Radiometer Suite (VIIRS) sensors on the Suomi-NPP satellite (SNPP) launched in November 2011, and the NOAA-20 satellite launched in November 2017. NOAA marine color data is processed using the NOAA Multi-Sensor Level 1 to Level 2 Processing System (MSL12). The system was developed by the NOAA/STAR Marine Colour Team. Further details on the datasets are available at https://coastwatch.noaa.gov/cw/satellite-data-products/ocean-color/near-real-



time/viirs-multi-sensor-gap-filled-chlorophyll-dineof.html. The data has been updated since 2018. In this study, the average
of the same month in the simulation period was selected for calculation.
As can be seen from Fig.2(e, f, g, h), the seasonality showed a trend of October > April > July > January. The reason of the
specific spatial and temporal distribution is that temperature gradually increased in April, and upwelling lead to the delivery
of nutrients to the at the ocean surface, which translates into plankton growth and corresponding increase in Chla. However,
with the further increase of temperature in July, fish, shrimp and other benthic organisms in the ocean gradually enter the
breeding period, and a large number of larvae feed on plankton, thus resulting in a decrease in Chla due to the increase in the
consumption of plankton. In October, the consumption of plankton by the larvae of marine organisms decreased. In addition,
a large amount of nutrients were accumulated in the sea water in July, and plankton began to multiply in large numbers.
Therefore, the highest Chla value appeared in autumn. Then in winter (Janurary), the temperature of sea water decreased and
the life activities of plankton weakened, and therefore Chla decreased. In addition, from the perspective of spatial
distribution, the seasonality of Chla in the offshore area is more obvious, while the seasonality in the pelagic area is not
obvious. High values of Chla are mainly distributed in the offshore areas, and also appear in the areas far from the land in the
East China Sea in spring. On the one hand, human activities discharge nutrients into the ocean from the estuaries of Liaohe
River, the Yellow River, Huaihe River, the Yangtze River and the Pearl River. In addition, the fishing grounds of Yellow-Bo
Sea, Zhou-Shan, south China sea coast and Beibu Gulf Fisheries, which distribute from north to south, lead to a coastal input
of increased nutrients, so plankton blooms in the coastal areas. On the other hand, at the estuary of rivers, the difference in
salinity between fresh water and sea water causes sea water to churn, hence, sea water turbidity increases, reducing the
intensity of light and restricting the growth of plankton. As the sea water moves outward, its vertical stability increases, the
sediment deposition and the transmittance increase, which results in phytoplankton multiplication in large numbers, and a
high value of Chla appears again in the pelagic area of the East China Sea (Zhang et al., 2016).
**2.5.3 Sea Surface Salinity**
The SSS data used for ion intensity (I) calculation started in April 2015, when the level-3 grid-based SSS daily mean data set
with spatial resolution of 0.25° was obtained from observation and inversion by SMAP (Soil Moisture Active Passive)
satellite. Coast Watch/ Ocean watch Level-3 SSS products are generated directly from the NASA Jet Propulsion Laboratory
(JPL) SMAP Level-2B SSS near real time SWATH HDF5 file. These products improve the application of satellite SSS
products, to reduce data delay to 24 hours (https://coastwatch.noaa.gov/cw/satellite-data-products/sea-surface-
salinity/smap.html). In this study, the data of April, July and October 2015 were adopted as the average of the research
period, while the missing data of January 2015 were replaced by the data of January 2016. As can be seen from Fig.2 (i, j, k,
l), the seasonality of SSS was April > July > January > October. This is mainly because the value of SSS can be affected by
precipitation, evaporation, fresh water input from outflow river and tidal action. From spring (April) to summer (July),
precipitation in the southeast coastal areas of China gradually increases, and the amount of water input from estuaries also
greatly increases, leading to a gradual decrease in SSS, which reaches the lowest value in autumn (October). Then in the





autumn, the precipitation begins to decrease, and SSS also begins to rise gradually, and reaches peak again in the spring of
the next year. The seasonality of SSS in the East China Sea is obvious, mainly because it is significantly affected by the
Yangtze River, and SSS is inversely proportional to the runoff input. However, in the South China Sea and east of Luzon
Island, the variation of SSS is not obvious, which is mainly related to the climate characteristics of high temperature and
rainy periods all year round in this region (Liang et al., 2019).
**2.5.4 Wind Speed**
Due to the lack of wind speed (WS) data with high temporal and spatial resolution, and issues of data integrity, the WS used
in this study was simulated by WRF based on FNL forecast data. Table S1 shows the comparison between the simulated
results and the observed values of meteorological elements. It can be seen from the results that the model can simulate the
numerical values and variation trends of WS, temperature and relative humidity.
It can be seen from Fig. 2 (m, n, o, p) that the wind speed is strongest in winter (January), weakest in summer (July) and
transitions from spring to summer, which is also consistent with satellite data reported in other studies (Lin et al., 2000). The
wind blows from the ocean to the land in summer, and from the land to the sea in spring, autumn and winter, which also
causes the land to be affected by more MBE in summer.
**3 Results and Discussion**
**3.1 Spatial and temporal distribution of methylamines emissions**
The emission fluxes of the three types of amines in January, April, July and October are shown in Table 5. Negative values
indicate that the amine is transported from the ocean to the atmosphere, where the ocean is the source of the amine. Positive
values indicate that amines are transported from the atmosphere to the ocean, which is a sink of amines. To facilitate the
comparison with terrestrial AE, the positive and negative signs were adjusted in this study. The positive values of the
original calculation results were changed to negative values, indicating that the ocean is the sink. A change in the negative
value of the original calculation to a positive value indicates that the ocean is the source. The established spatial and
temporal distribution of the amines emissions is shown in Fig. 3, and the diurnal variation of the emission flux is shown in
Fig. 4. As we can see, the MBE fluxes of amines are quite substantial. The MBE emission fluxes of MMA are close to
terrestrial AE in January and October, but is relatively low compared with terrestrial AE in April and July. TMA was higher
in January and October, but lower in April and July. Overall, TMA was significantly higher than terrestrial AE. The peak
value of terrestrial AE is concentrated between 4:00 and 8:00. Due to the relative stability of the marine environment, the
diurnal variation range of MBE is not as obvious as that of terrestrial AE, but it can still be seen from Fig. 4 that the MBE
have a slight increase in the daytime.
The influence of the four variables on the exchange fluxes is shown in Fig. 5. As shown in Eq. (4), WS is directly used in the
calculation of ocean-atmosphere exchange coefficient $K_g$, which is directly proportional to $K_g$, and has the most direct





influence on the calculation of exchange fluxes. Higher WS will accelerate the material exchange between the ocean and the atmosphere, which is directly reflected in the change of the exchange fluxes, and the two show a significant linear relationship (Fig. 5(a)). For MMA and TMA, the ocean is the source, and the increase of WS will lead to a linear increase in the emission fluxes. For DMA, the ocean is a sink, and the increase in WS also accelerates the flow of atmospheric DMA into the ocean. The other three variables have an indirect effect on the exchange fluxes mainly by affecting the calculation of intermediate variables.

With the increase of Chla, the direction of amines exchanges between the ocean and atmosphere showed a trend of transferring from the atmosphere to the ocean (Fig. 5 (b)). According to Eq. (9), the increase of Chla will lead to the decrease of pH, hence, it is not conducive to the overflow of amines. Chla is used to indicate primary production in the water body, and high Chla indicates a significant increase in phytoplankton in the water body. Water eutrophication is caused by the massive growth of phytoplankton due to the continuous importation of anthropogenic nutrients into the sea by rivers. Thus, the increased organic matter is transported to the subsurface water by settling and being decomposed by microorganisms. This process consumes oxygen in the water and forms a hypoxic environment. With the mixing of the water, the pH of the water changes. Zhao et al. (2020), based on the observation data of summer voyage from the Pearl River Estuary to the northern continental shelf of the South China Sea, found that the water on the west side of the Pearl River Estuary with obvious mixing of fresh water and seawater is characterized by low dissolved inorganic carbon (DIC) and high pH, while for the area with 20-30 meters water depth outside the mixing area, it is characterized by high DIC and low pH. The main source of amines comes from the degradation of organic matter in sediments, therefore, an increase in Chla might mean more acidification in the ocean, making amines more soluble in seawater.

An increase in SST will also lead to a decrease of the fluxes of amines discharged from the ocean to the atmosphere (Fig. 5 (c)). As can be seen from Eq. (9), the increase of SST will lead to the decrease of pH, which is not conducive to the overflow of amines. Due to the high stability of the marine environment, the variation range of SST itself is small, hence, SST has the least influence on the exchange flux among the four elements. It should be noted that, El Niño conditions occurred by late May in 2015, which increased the global average temperature and affected the weather patterns in the study area (Kennedy et al., 2016). The annual average SST of our study areas was 0.5-1.0 °C higher than the average value recorded during the period from 1961 to 1990 (Kennedy et al., 2016). Figure 5(c) shows that the exchange fluxes of amine are positively correlated with SST, but the sensitivity is low.

It can be seen from Fig. 5 (d) that the increase of SSS will increase the tendency of the amines to overflow the water surface. It can be inferred from Eq. (9), (10), (11), and (12) that the increase of SSS will further inhibit the ionization of amine in seawater, which makes it more prone to overflow from the water surface, resulting in an increase of the exchange fluxes. However, in general, WS and Chla are the main factors affecting the exchange fluxes.





## 3.2 Spatial and temporal distribution of methylamine concentrations

Due to the temporal and spatial resolution of the model and the incompleteness of observation data, it is not possible to
directly compare the model simulation results with the observation data in this study. But on the whole, the simulated results
of amines are close to their actual concentrations in the atmosphere in terms of magnitude, and inclusion of marine biogenic
emissions can make up for the shortcomings of the previous models' underestimation of amines. Table 6 summarizes the
amine concentrations for this study and data from other published papers (both simulation and measurement data).
Considering the large simulation area and low grid resolution in this study, the Yangtze River Delta region with relatively
dense observation sites was selected to verify the calculation results effectively. Without considering the contribution of
MBE, the simulation results in this study accord with the general rule of DMA>MMA>TMA, which can be attributed to the
difference in AE fluxes. Previous simulations often underestimated the amines concentration (Mao et al., 2018). The
contribution of MBE to TMA and MMA will effectively improve the regional simulation value. Moreover, due to the
increased consumption of ·OH by TMA and MMA, the consumption of DMA will decrease and the average concentration
will increase slightly in the spring and summer when the photochemical reaction is strong. However, there was no significant
change in the range in autumn and winter (Table 6). Figures 6, 7, and 8 show the spatial and temporal distribution of the
simulated concentrations of the three amines in the presence and absence of MBE. Due to the contribution of MBE, the
ambient concentration of TMA of coastal areas increased multiple times, and the concentration of amines above the offshore
area also increased significantly. The specific amount of increase is shown in Table 6. In January, April, July and October,
TMA increased by 37.1%, 23.7%, 36.5% and 27.9%, respectively, while TMA increased by more than 10% in the area over
500 km from the coastline. The increase rates of TMA in the four months even reached 229.6%, 107.1%, 116.7% and
171.5%, with significant increases over Bohai Bay, Yangtze River Delta, Leizhou Peninsula and the Sea of Japan. The
second is MMA, with an average increase rate of 4.0%, 3.7%, 7.8% and 3.6% in January, April, July and October,
respectively. The increase of MBE mainly affects the area about 50 kilometers from the coastline. In the four months, the
average increase rates of MMA in this part reached 57.9%, 23.9%, 26.2% and 37.5%, with considerable increases over
Bohai Bay, Yangtze River Delta, the waters around Leizhou Peninsula and the Sea of Japan. Moreover, due to the obvious
increase of the other two amines, ·OH decreased and the consumption of DMA decreased correspondingly, thus leading to a
small increase in the DMA concentration. The seasonality of MMA and TMA increases is consistent with the seasonality of
emission fluxes (January > October> April > July).
As can be seen from the above description, WS could be the most important factor affecting MBE fluxes because it is
directly used to calculate the air-ocean exchange coefficient ($K_g$) (Eq. (4)). At the same time, wind direction is another
important factor that affects MBE. In July, the monsoon blows from the sea to the land, causing the effects of MBE to reach
deep into China's interior. It can also be seen from Figures 6, 7 and 8 that the concentration of amine over land in July
changed the most. Furthermore, the influence of MBE is also conditioned by the monsoon and the topography of China.
Specifically, the terrain of China can be divided into three steps: the low east, the high west and the third step extending to





the ocean. The third step, i.e., the offshore continental shelf, is usually the region less than 500 meters above sea level, and is
most affected by MBE. Moreover, the influence of MBE on western China is weakened due to the barrier between the
second and third steps (Mt. Greater Hinggan–Mt. Taihang–Mt. Wushan–Mt. Xuefeng). Therefore, the prevailing southwest
monsoon in July can smoothly drive the marine air mass into the third step area, and its influence area is wider due to the
lack of obstacles in the north-south direction. However, in January, April and October, the prevailing northerly monsoon
brings more terrestrial air, limiting the contribution of MBE. The geographical location also has an important influence on
the MBE contribution. Some regions are always affected by the strong ocean air mass. For example, Hainan Island and the
Shandong Peninsula are surrounded by the sea on many sides, hence, they are affected by strong oceanic air mass during all
four seasons. Moreover, Southern China (especially, the coastal areas of Fujian and Guangdong provinces) is more affected
by MBE than northern China due to the prevailing winds blowing from the sea to the land.
Moreover, it should be noted that after the addition of MBE, not all regions showed an increase in the concentration of
amines; some regions showed a small decline. However, compared with the increase, the decrease was not significant. Even
in July, when the ocean source had the greatest influence, the average concentration in the drop area was lower than 10%
(Table 7). This situation may be caused by the reaction equilibrium of ·OH and NOx in the atmosphere which is affected by
MBE. If MBE are added, ·OH consumption will take place, and photolysis of HONO will be promoted to generate more NO.
However, NO will react with ·OH$_2$ at a faster reaction rate to generate ·OH, which will lead to an increase in ·OH
concentration in some areas. In the upwind region, the amine is consumed and cannot be replenished enough, which leads to
the decrease of the amine concentration in some areas after the increase of MBE.

### 3.3 The main factors affecting amines simulation

The simulation of amines is mainly influenced by two factors, namely the emissions and model mechanism. Therefore, this
study designed sensitivity tests from the two aspects, and discussed the contribution of different types of terrestrial AE to
amines, the data used to calculate MBE and the influence of wet deposition on the simulation of amines.

### 3.3.1 Terrestrial anthropogenic emissions

The contributions of different terrestrial AE to amines concentration were significantly different. As shown in Table S2,
agricultural emission is the major source of the three types of amines. However, the agricultural emissions of DMA are
relatively low compared with the agricultural emissions of MMA and TMA accounting for more than 80%. Residential
emission is another important source of DMA, accounting for more than 10% in each of the four months, and even reaching
36.5% in January. Industrial emission is the third major source of amines, accounting for about 5%. The contribution of
transportation emissions is very low (<1%), and the contribution of power plant emissions is negligible. In addition, the
contribution ratio of different emissions also presents a strong seasonality. Due to the wide distribution of agricultural
emissions and the absolute dominance among all kinds of emissions, this study only conducted sensitivity tests on residential





emissions and industrial emissions, i.e., quantitative analysis was conducted on the change of amines concentration after the complete removal of a certain type of emissions.

Fluxes and spatial distribution of different emissions will affect the amines concentration. After the reduction of residential emissions, DMA showed stronger sensitivity than the other two amines due to residential emissions' high proportion in the total emission of DMA. After the reduction of industrial emissions, the changes of the three amines were not different from each other and were consistent with the reduction of emissions (~5%). However, the spatial distribution of residential and industrial emissions make the change of amines simulated concentration significantly different (Fig. S1). As can be seen from Fig. S2 and S3, residential emissions are distributed in a wide range. Although like agricultural emissions, they have relatively high fluxes in areas such as Sichuan Basin, North China Plain, Yangtze River Delta and Pearl River Delta, the difference is not obvious. Therefore, the decrease range of amines concentration is approximately uniform in the simulated region. The industrial emissions are concentrated in the north of the Yangtze River, especially in Shandong, Shanxi, Shaanxi, Henan, Hebei, etc., and there is a significant difference between the high value area and the low value area. Therefore, although the proportion of the total industrial emissions are not high, the local decline caused by the reduction is relatively high, and the influence is greater mainly in the places where industrial emissions distribution is more concentrated.

### 3.3.2 Marine biological emissions

To further explore the influence of satellite data on the spatial distribution of the simulated amines concentration, two variables, Chla and WS, which have obvious effects on the emission flux of amines, were selected and sensitivity tests were conducted by increasing or decreasing the values by 50%. SSS and SST were not considered due to their small variation range and small influence on the amines exchange fluxes.

The concentration change of Chla is related to the calculation of pH, hence, it has an indirect effect on the calculation of the emission flux of amines. When Chla decreased, the pH of seawater would rise to a certain extent, which would promote the overflow of amines from seawater. With the increase of Chla, the pH of seawater will show a downward trend, which will inhibit the overflow of amines. After increasing or decreasing Chla, the maximum variation ranges of seawater pH in January, April, July and October were ±0.27, ±0.52, ±0.56 and ±0.70, respectively. The higher the concentration of Chla, the more obvious was the pH change caused by it, which was mostly concentrated in the coastal areas. In addition, it can be seen from the discussion in Section 3.1 that the influence of Chla on emission fluxes gradually decreases with the increase of Chla and eventually tends to stabilize. Therefore, the effect of Chla reduction on emission fluxes is higher than that of Chla increase, as can be clearly seen from Table S3. Moreover, the effect of Chla on MMA is significantly higher than that of TMA, mainly because MMA is more alkaline, thereby making it more sensitive to changes in pH.

From Fig. S4 and Fig. S5, we can see the spatial and temporal changes of the final simulated values of amines after reducing and increasing 50% Chla. The influence of Chla variation is mostly concentrated in the maritime area and the area adjacent to the coastline (<27km), and the magnitude of the influence is directly related to the variation range of amines emissions.



Except for July, the impact on land is small, and the fluctuation is generally less than 5%. In July, due to prevailing winds
blowing from the ocean to the land, the amine variation is greater (>5%). In addition, although the change of MBE of DMA
is not considered here, the changes of MMA and TMA have an impact on the reaction cycle of ·OH, which indirectly affects
the concentration of DMA. The range of DMA over land is like that of TMA and MMA, but the effect over the ocean is
mainly downwind of the prevailing wind direction. This phenomenon is mainly caused by the fact that the reactivity of DMA
is higher than that of TMA and MMA; therefore, DMA reacts with ·OH faster than the other two amines in the upwind
direction, and its concentration is less affected. However, as time goes on, the effect of TMA and MMA concentration
changes becomes prominent, and DMA changes greatly downwind of the prevailing wind direction.
There is a positive linear correlation between WS and the emission fluxes. Thus, a 50% reduction or increase in WS is
equivalent to a 50% reduction or increase in the overall MBE. Similar to the results obtained from changing Chla, changing
WS mainly had a significant effect on the concentration of amines over the sea, and the effect on land was mainly confined
to the coastal areas. In the example of the 50% reduction of WS (Fig. S6), for MMA and TMA, the 50% reduction of MBE
resulted in significant changes in the simulated values of both over the ocean. Moreover, due to the prevailing wind blowing
from land to sea in January, April and October, the changes of these two amines over the land were not obvious. In July, due
to the prevailing wind blowing from the ocean to the land, the change of MBE has a greater impact on the land. However,
there are still differences between the two. As the MBE fluxes of MMA are slightly lower, the amine concentration in the
offshore area near the land can be supplemented by AE, so the reduction of the amine concentration in the offshore area is
relatively small, and it is obvious in January, April and October. However, due to the higher fluxes, TMA is more sensitive
than MMA, and the supplementary effect of terrestrial AE on TMA over the offshore area is not obvious.
**3.3.3 Wet Deposition**
Like ammonia, amines are considerably soluble in water, and hence the influence of wet deposition on the simulated
concentration of amines cannot be neglected. As seen from Fig. S7, the cumulative precipitation presents great differences in
different simulation periods. In the simulation period of January, April and October, the precipitation was low, going below
50 mm in October. However, in July, there is significant precipitation and an obvious rainfall zone between Jiaodong
Peninsula and Guangdong province. This temporal and spatial distribution must have different effects on the concentration
of amines. To explore the influence of wet deposition on the simulation of amines, the apparent Henry constant of amines in
the model was set as 0.75 times and 1.25 times of the original, respectively. The larger the apparent Henry coefficient, the
easier it is for the amine to dissolve in precipitation and be removed from the atmosphere.
It was found that with the decrease of the apparent Henry coefficient, the amines became more difficult to dissolve in water,
and therefore, the concentration of amines increased in most areas (Fig. S8). However, in January, April and October, the
variation of amine concentration over land during the simulation period was generally small due to lower precipitation.
DMA is only affected by terrestrial AE, and when its concentration over the land increases, it is carried to the atmosphere
over the sea under the action of prevailing wind. As a result, the variation range of its concentration over the sea surface is





significantly higher than that of the other two amines, and the affected area is mainly concentrated in the downwind area of
the prevailing wind. In July, due to more precipitation, the effect on amines was significantly stronger than that in other
months. In addition, the area with more obvious change coincides with the rainfall zone mentioned above. Moreover, the
transport of prevailing winds from the ocean to the land and the blocking effect of the mountains dividing steps two and
three in the east made the whole eastern coastal area exhibit the most obvious change in the concentration of amines. When
the Henry's coefficient is increased, the trend is reversed.
**4 Conclusions**
In this study, numerical modelling was employed to calculate amine emissions from ocean sources by using satellite data
such as SST, Chla, and SSS. Algorithms were used to establish a more reasonable MBE contribution for the amines
including $C_1$-amine ($CH_5N$, MMA), $C_2$-amines ($C_2H_7N$, DMA) and $C_3$-amines ($C_3H_9N$, TMA). The ocean acts as the source
for MMA and TMA in all seasons and acts as a sink for DMA. However, if the traditional ratio method is adopted, whether
the ocean is the source of amine emission depends entirely on whether ammonia can be discharged from the ocean. This
"one-size-fits all" method is not consistent with the actual situation, and further affects the simulation results of subsequent
models. The satellite data used in this study can better reflect the exchange direction of amines between ocean and
atmosphere and reflect the emission characteristics of different amines, which is more consistent with the reality.
The calculated spatial and temporal distribution of methylamine revealed that MBE can effectively increase the amines
concentrations in most areas. Due to the contribution of MBE, the ambient concentration of TMA of coastal areas increased
multiple times, and the concentration of amines above the offshore area also increased significantly. TMA increased by more
than 10% in the area over 500 km from the coastline. The increase rates of TMA in four months reached 229.6%, 107.1%,
116.7% and 171.5%. The next largest increase was observed for MMA, with an average increase rate of 4.0%, 3.7%, 7.8%
and 3.6% in January, April, July and October, respectively. The increase of MBE mainly affects the area about 50 kilometers
from the coastline. In the four months, the average increase rates of MMA in this part reached 57.9%, 23.9%, 26.2% and
37.5%. WS and Chla were found to be the dominant factors affecting MBE fluxes. WS is directly used in the calculation of
the air-ocean exchange coefficient ($K_g$), and the influence of MBE is conditioned by the monsoon and the topography of
China. The prevailing southwest monsoon in July can thrust the marine air mass into the third step area (the offshore
continental shelf), and its influence area is wider due to the lack of obstacles in the north-south direction. However, in
January, April and October, the prevailing northerly monsoon brings more terrestrial air, limiting the influence of MBE. The
special geographical characteristics of some areas, e.g., Hainan Island, Fujian and Guangdong provinces etc., make them
more prone to the influence of the strong ocean air mass. Chla indirectly influences the calculation results of exchange flux
by affecting the calculation of pH. With the increase of Chla, exchange flux decreases sharply at first and then becomes
constant, and the ocean also transforms from a source of amines to a sink. In addition, the emission flux and spatial
distribution of AE, and wet deposition also affect the simulation of amines.



On the other hand, it should be noted that due to the limitation of the algorithm (e.g., Eq. (8)), the MBE established in this
study is only applicable to the North Pacific region. Additionally, the dependence on satellite data and corresponding
accuracy makes MBE uncertain, and the amount of data processing required also increases the difficulty and cost of
establishing an emission source inventory. Therefore, further studies are needed to establish a more accurate numerical
simulation for the amines.
**Author contribution**
**Competing interests**
The authors declare that they have no conflict of interest.
**Financial Support**
This work was funded by the National Natural Science Foundation of China (grant numbers 42121004, 41905107, 41905086,
41425020), Guangdong Innovative and Entrepreneurial Research Team Program (grant number 2016ZT06N263), and the
Special Fund Project for Science and Technology Innovation Strategy of Guangdong Province (grant number
2019B121205004), the Natural Science Foundation of Guangdong Province (grant number 2021A1515011248), Science and
Technology Projects in Guangzhou (grant number 202102080141), Natural Science Foundation of Tianjin City (grant
number 21YFSNSN00200), the AirQuip (High-resolution Air Quality Information for Policy) Project funded by the
Research Council of Norway, the Collaborative Innovation Center of Climate Change, Jiangsu Province, China, and the
high-performance computing platform of Jinan University.

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



Table 1 Amines-to-ammonia mass emission ratios (Mao et al., 2018)

|  | Transportation Emissions | Residential Emissions | Other industrial Emissions | Chemical industrial Emissions (Power Plant) | Agricultural Emissions |
|---|---|---|---|---|---|
| MA | 0.0011 | 0.0011 | 0.0015 | 0.026 | 0.0011 |
| DMA | 0.00090 | 0.010 | 0.0018 | 0.0070 | 0.0015 |
| TMA | 0.00040 | 0.00060 | 0.00050 | 0.00040 | 0.00040 |




Table 2 Configuration of WRF-Chem parameterization schemes

| Physical parameterization schemes | | Chemical parameterization schemes | |
|---|---|---|---|
| mp_physics=11 | CAM5 Morrison-Gettleman scheme to be used with MAM chemistry | chem_opt=503 | CBMZ with CAM-MAM3_AQ |
| bl_pbl_physics = 9 | CAM5 UW PBL scheme to be used with CAM-MAM chemistry | emiss_inpt_opt=104 | Carbon Bond 4-emission speciation adapted after reading the RADM2 data file. Secondary Organic Aerosol (SOA) precursors computed from input data as well. Using for CAM5 microphysics and MAM 3-mode aerosol |
| cu_physics = 7 | CAM5 Zhang-McFarlane scheme to be used with MAM chemistry | emiss_inpt_opt=9 | Converting default RADM2 gas emissions to CBMZ. Aerosol emissions are speciated to MAM 3-mode aerosols |
| | | cam_mam_mode = 3 | Number of MAM aerosol modes |
| | | cam_mam_nspec = 88 (original value: 85) | Number of MAM 3-bin aerosol species |






Table 3. A brief summary of previous research results on the distribution of amines.

| Location | Site Type | Period | [NH₃(g)] (pmol m⁻³) | [MMA(g)] (pmol m⁻³) | [DMA(g)] (pmol m⁻³) | [TMA(g)] (pmol m⁻³) | [NH₃(s) tot⁺] (μmol m⁻³) | [MMA(s) tot⁺] (μmol m⁻³) | [DMA(s) tot⁺] (μmol m⁻³) | [TMA(s) tot⁺] (μmol m⁻³) | Ref. |
|---|---|---|---|---|---|---|---|---|---|---|---|
| Oahu, Hawaii | Coast | July–August 1985 | - | 11±5 | 93±51 | 30±19 | - | 52±20 | 1.5±20 | 12±3 | |
| Narragansett, Rhode Island | Coast | | - | 52±12 | 240±40 | 100±40 | - | | | | Van Neste et al. (1987) |
| Vineyard S. | Coast | | - | | | | - | 200±58 | 8.9±4.4 | 41±27 | |
| | | | | | | | | 32±5 | 8.9±1.1 | 10±13 | |
| Salt Pond | Coast | | | | | | | 55±20 | 6.7±2.2 | 10±4 | |
| Arabian Sea | Arabian Sea | 27 August–4 October 1994 | 946 | 114 | 39 | 0.9 | 108 | 16.2 | 4.54 | 0.044 | |
| Arabian Sea | Arabian Sea | 16 November - 19 December 1994 | 3780 | 143 | 196 | 8.1 | 156 | 25.6 | 4.85 | 0.14 | Gibb et al. (1999) |




Table 4 Physicochemical parameters used in flux calculations.

| Variable (Units) | Species | | | References |
|---|---|---|---|---|
| | MMA | DMA | TMA | |
| MW (g mol$^{-1}$) | 31.06 | 45.12 | 59.11 | |
| Henry's law constant, H (T=20 °C, I=0 mol dm$^{-3}$) | 0.0015 | 0.0013 | 0.0023 | Gibb et al. (1999) |
| Thermodynamic stability constant, p$K_{w\alpha}$ (T=20 °C, I=0 mol dm$^{-3}$) | 10.64 | 10.77 | 9.80 | |
| $K_H$ (M atm$^{-1}$) | 23.80 | 27.47 | 15.53 | |






Table 5 Emission fluxes of the three types of amines in January, April, July and October.

| Site | Date | MMA (pmol m$^{-2}$ s$^{-1}$) | DMA (pmol m$^{-2}$ s$^{-1}$) | TMA (pmol m$^{-2}$ s$^{-1}$) | Ref. |
|---|---|---|---|---|---|
| Eastern China | Jan. 2015 | -1.7±0.37 | 0.85±0.15 | -4.9±1.0 | This study |
| | Apr. 2015 | -1.3±0.41 | 0.70±0.19 | -4.0±2.1 | |
| | Jul. 2015 | -1.1±0.35 | 0.58±0.19 | -3.3±1.0 | |
| | Oct. 2015 | -1.4±0.36 | 0.74±0.16 | -4.2±1.0 | |
| Coastal Hawaii and Massachusetts | - | -1.8- -0.11 | 0.46-0.49 | -3.2 - -0.20 | Van Neste et al. (1987) |




Table 6 Comparison of gaseous methylamines from simulations and measurement results in different locations.

| Location (Site Type) | Data Type | Season | MMA (pptv) | DMA (pptv) | TMA (pptv) | Ref. |
|---|---|---|---|---|---|---|
| Nanjing, China (Industrialized) | Measured | Summer | 0.1-18.9 | 0.1-29.9 | 0.1-9.3 | Zheng et al. (2015) |
| Shanghai, China (Urban) | Measured | Summer | 15.7±5.9 | 40.0±14.3 | 1.1±0.6 | Yao et al. (2016) |
| Shanghai, China (Urban) | Measured | Spring | - | 4.7-26.1 | - | Yao et al. (2018) |
| | | Summer | - | 27.1-81.7 | - | |
| | | Autumn | - | 19.9-27.3 | - | |
| | | Winter | - | 3.5-16.7 | - | |
| Shanghai, China (Urban) | Measured | Summer | 15.71 | 40.20 | 1.13 | Mao et al. (2018) |
| | Simulated | | 4.97±6.61 | 16.33±25.15 | 1.01±1.37 | |
| Nanjing, China (Urban) | Measured | Summer | 4.35 | 7.08 | 1.91 | |
| | Simulated | | 6.39 | 10.56 | 1.12 | |
| Nanling Mountain, southern China (Background) | Measured | Summer | 51.9±58.0 | 111.3±82.3 | - | Liu et al. (2018) |
| | | Autumn | 56.1±32.4 | 45.5±34.7 | - | |
| Bohai Sea and Yellow Sea (Marine) | Measured | Winter | - | 0.006 ± 0.006 µg m⁻³ | 12±4 | Gao et al. (2022) |
| The Yellow Sea (Marine) | Measured | Winter | - | 0.002±0.011 µg m⁻³ | 14±5 | |
| East China Sea (Marine) | Measured | Winter | - | 0.012±0.011 µg m⁻³ | 8-91 (26 ± 17) | |
| The Yellow Sea and Bohai Sea (Marine) | Measured | Winter | - | 0.018±0.021 µg m⁻³ | 0.5 ng m⁻³ | Chen et al. (2021) |
| the coastline of eastern China (Marine) | Measured | Spirng | - | 11 ± 6.5 ng m⁻³ | 5.4 ± 2.4 ng m⁻³ | Chen et al. (2022) |
| The Yangtze River Delta, China (Urban) | Simulated (Without MBE) | Spring | 5.15±3.79 | 5.83±4.42 | 1.03±0.75 | This study |
| | | Summer | 6.79±4.27 | 7.07±4.41 | 1.34±0.83 | |
| | | Autumn | 18.01±8.76 | 21.85±10.94 | 3.59±1.75 | |
| | | Winter | 10.64±5.94 | 17.82±10.90 | 2.24±1.25 | |
| | Simulated (With MBE) | Spring | 5.22±3.78 | 5.84±4.41 | 1.57±1.07 | |
| | | Summer | 7.25±4.56 | 7.51±4.71 | 2.50±1.60 | |
| | | Autumn | 18.16±8.77 | 21.99±10.96 | 5.03±2.22 | |
| | | Winter | 10.67±5.95 | 17.71±10.94 | 2.72±1.51 | |




Table 7 Changes in methylamine concentrations with MBE contribution (Unit: %).

| | MMA | | | DMA | | | TMA | | |
|---|---|---|---|---|---|---|---|---|---|
| | MD[a] | MI[b] | MA[c] | MD[a] | MI[b] | MA[c] | MD[a] | MI[b] | MA[c] |
| Jan | -1.2 | 4.0 | 2.7 | -1.4 | 1.5 | 1.4 | -1.1 | 37.1 | 22.9 |
| Apr | -1.9 | 3.7 | 2.8 | -2.1 | 2.6 | 2.3 | -1.8 | 23.7 | 14.1 |
| Jul | -3.9 | 7.8 | 5.9 | -4.1 | 6.2 | 5.1 | -4.0 | 36.5 | 22.0 |
| Oct | -2.3 | 3.6 | 7.5 | -2.4 | 2.2 | 2.3 | -2.5 | 27.9 | 16.7 |

[a] The mean amplitude of the region where the concentration decreased.
[b] The mean amplitude of the region where the concentration increased.
[c] Mean change in absolute concentration.


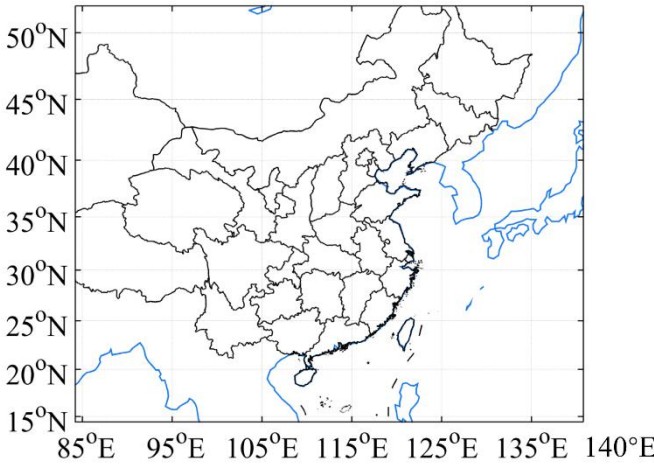


**Figure 1: Simulation domain in this study.**



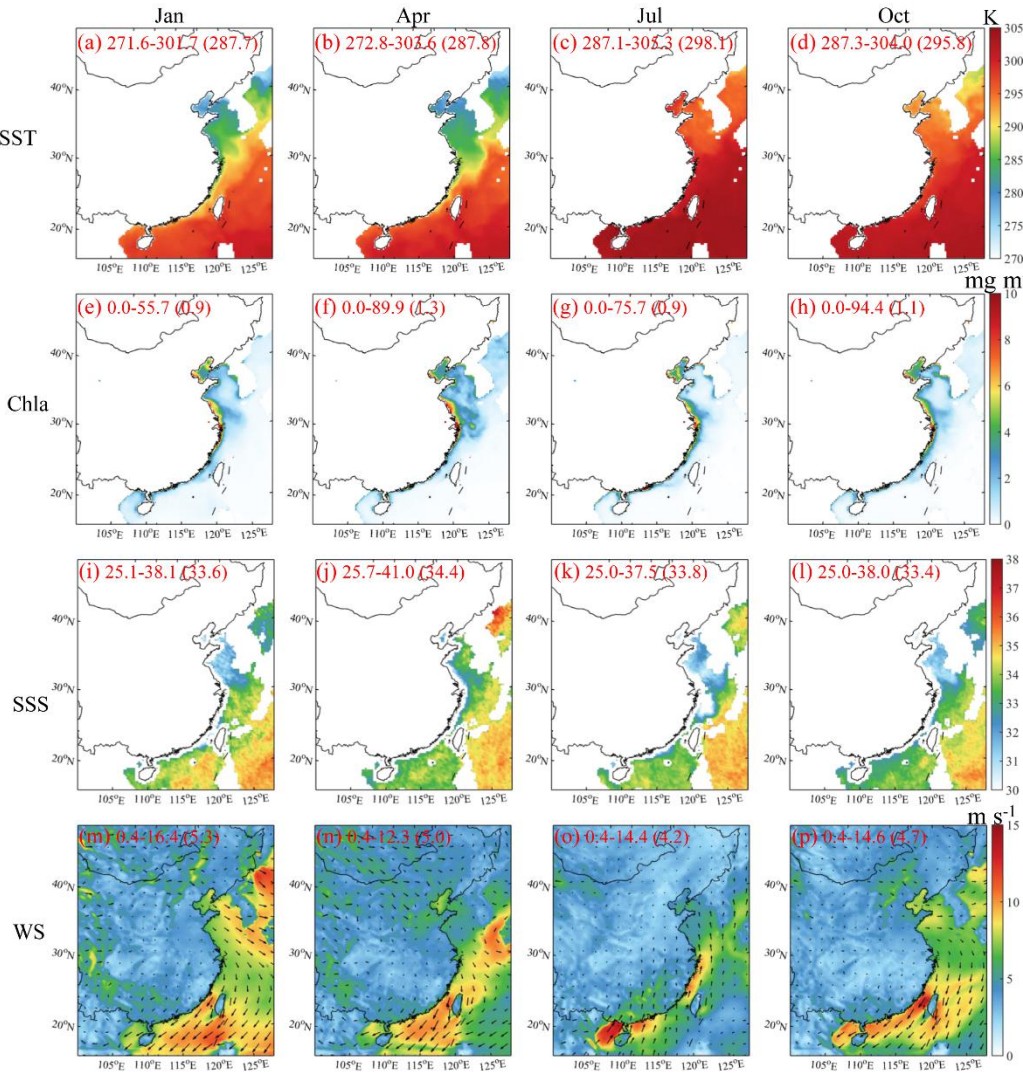

**Figure 2: Temporal and spatial distribution of SST (a-d), Chla (e-h), SSS (i-l), and WS (m-p) in January, April, July, and October. The numbers marked next to the serial number are the minimum, maximum, and average values from left to right.**




**Figure 3: Spatial and temporal distribution of methylamines emissions: (a, e, h, k) MMA; (b, f, I, l) DMA; (c, g, j, m)TMA.**



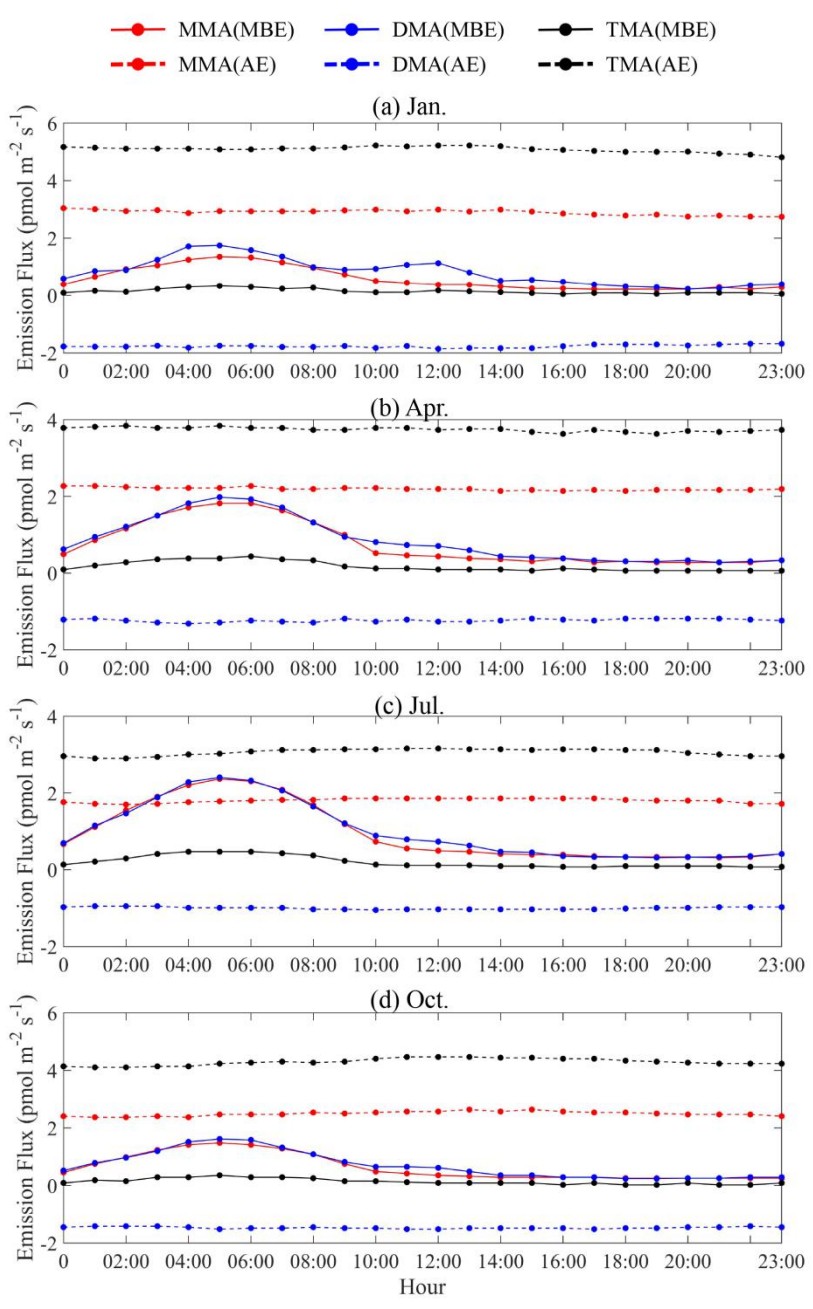


**Figure 4: Diurnal variation of emission fluxes of methylamines. (a) Jan. (b) Apr. (c) Jul. (d) Oct. The solid lines represent the**
**contribution from AE, the dotted lines represent the contribution from MBE.**



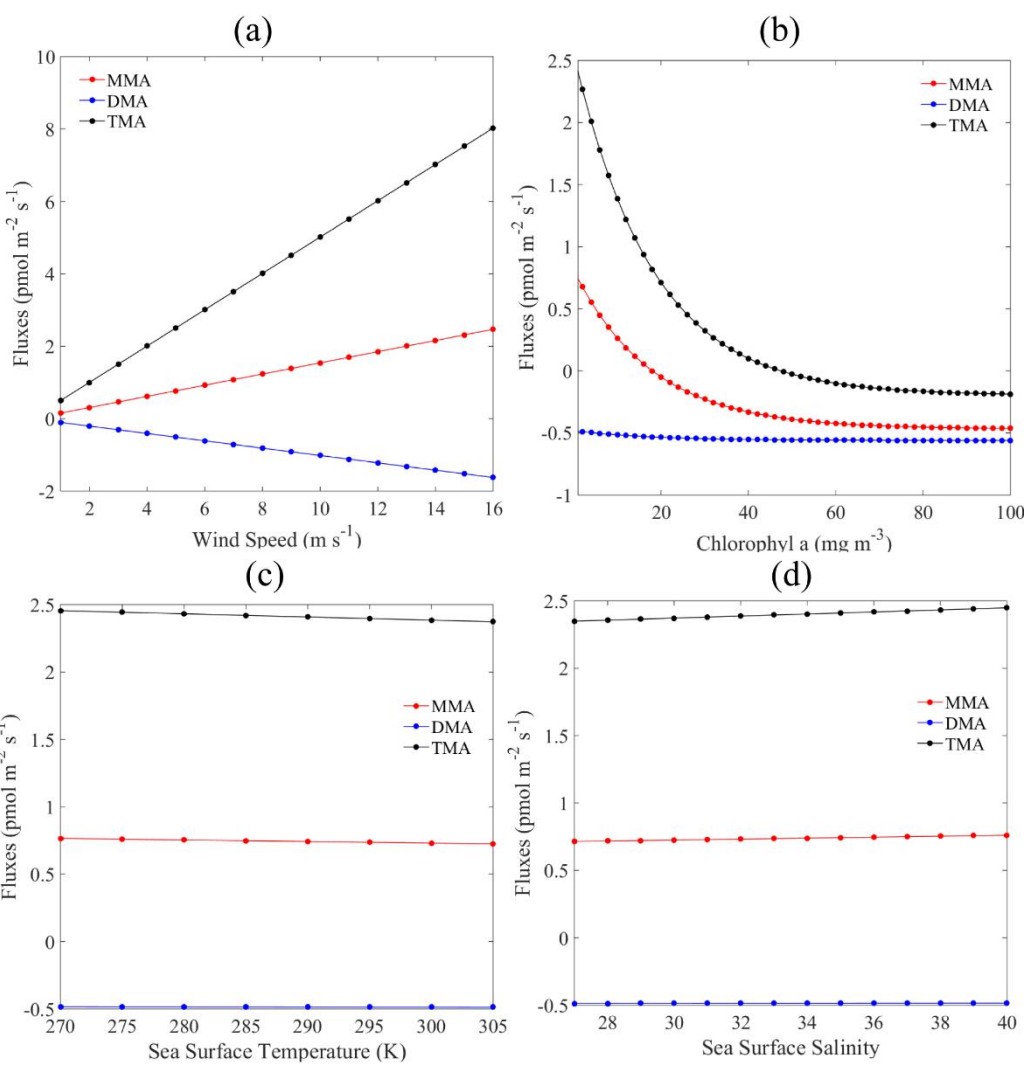


**Figure 5: Influence of WS, Chla, SSS and SST on amines exchange fluxes.**



**Figure 6: Spatial and temporal distribution of MMA simulated concentration. (a) (e) (h) (k): Without MBE (Unit: pptv). (b) (f) (i) (l): With MBE (Unit: pptv). (c) (g) (j) (m): MMA concentration changes on land (Unit: %). The solid red line represents the boundary between the second and third steps of the Chinese terrain.**



722

**Figure 7: Spatial and temporal distribution of DMA simulated concentration. (a) (e) (h) (k): Without MBE (Unit: pptv). (b) (f) (i) (l): With MBE (Unit: pptv). (c) (g) (j) (m): DMA concentration changes on land (Unit: %). The solid red line represents the boundary between the second and third steps of the Chinese terrain.**



**Figure 8: Spatial and temporal distribution of TMA simulated concentration. (a) (e) (h) (k): Without MBE (Unit: pptv). (b) (f) (i) (l): With MBE (Unit: pptv). (c) (g) (j) (m): TMA concentration changes on land (Unit: %). The solid red line represents the boundary between the second and third steps of the Chinese terrain.**