# Peer review of "Contribution of marine biological emissions to gaseous methylamines in the atmosphere: an emission inventory based on satellite data"

_Atmospheric Chemistry and Physics, 2022_

## Author Comment (AC1)

**Manuscript id: acp-2022-394**

**Manuscript title:** Contribution of marine biological emissions to gaseous methylamines in the atmosphere: an emission inventory based on satellite data

**Responses to Reviewers' comments:**

We thank the Reviewer for the constructive suggestions and helpful comments. We provide below itemized responses to each of the Reviewer's comments. The comments are given in bold while responses are in normal font. Changes made to the manuscript are shown in blue.

**Reviewer #1:**

In the current study the authors provide a combined satellite and model study to estimate oceanic emissions of methylamine (MMA), dimethylamine (DMA) and trimethylamine (TMA) along the Chinese coastlines. Therefore, sea surface temperature (SST), chlorophylla (chl-a), sea surface salinity (SSS) and wind speed data were used. Recent investigations show that satellite data are a useful tool to simulate and understand emissions from the ocean. The study reveals that amine emissions from the ocean can have significant contribution to gas-phase TMA and MMA concentrations, but not to DMA concentrations.

Through sensitivity studies wind speed and chl-a concentrations were found to be important drivers of amine emissions. The modeled gas-phase concentrations of MMA, DMA and TMA are compared with measurements in that region and found good matches. Regarding the importance of amines for new particle formation and current limitations the paper addresses relevant scientific questions in the field of atmospheric chemistry.

Nevertheless, emissions from the oceans towards the atmosphere require good established concentration measurements within the sea water. These are not given for the Chinese coastline and thus the authors used an average value derived from different measurements. Here, high uncertainties can exist. Furthermore, the authors do not use established measured physical and chemical parameters for these amines. Besides, it seems that there is a bug in the calculation of the pKa value. Therefore, the simulations have to be reperformed.

**Response:**

According to the suggestions of reviewer, we have reperformed the simulations with corrected parameters and revised the whole manuscript accordingly. We mainly modified the calculation methods of the Henry's Law constant and pKa, and have added satellite-derived NH3 column concentration to obtain a more correct and rigorous algorithm. Based on the above modification to the original algorithm, the new results show that the flux of amines from the atmosphere to the ocean slightly increases, while the flux from the ocean to the atmosphere slightly decreases; however, the main conclusions of the manuscript does not change.

In addition, we have obtained observational data from marine site for verification of the simulation results, thereby improving the simulation performance of amines concentrations at different types of sites (urban and marine sites) with the addition of MBE.

**The paper needs major revision before publication.**

Main Comments:

1) Parameters such as pKa and Henry's Law coefficient are important to calculate the amine flux into the atmosphere. The authors use Henry's Law coefficients obtained for NH3 as it was done in recent studies. This approach is feasible if such values are not given in literature. However, these values are determined. Why are the authors not using the Henry's Law coefficients for MMA, DMA and TMA as provided in Sander (2015) and Leng et al. (2015)? The pKa values used in table 4 from Gibb et al. (1999) are valid for 25°C not 20°C. Why are the temperature dependent pKa values given in the review of Ge et al. (2011b) not used? The simulations have to be reperformed by using the values for MMA, DMA, and TMA.

**Response:**

We thank the reviewer for the constructive comments. To clarify, we actually used pKa and Henry's Law coefficient for MMA, DMA, and TMA instead of NH3 in our original manuscript but the temperature and ionic strength dependence of those parameters were adopted from NH3. In the revised manuscript, the temperature dependence of pKa and Henry's Law coefficient have been updated using the values for amines. However, when calculating the ionic strength dependence of pKa we used the relation for NH3 (Equation 9 in the revised manuscript) because such relation for amines is not available yet. The details of the revision are described below.

(1) Firstly, we have updated the calculation of the Henry's Law and associated constants. The new algorithm calculates the temperature-dependent Henry's Law coefficients based on the molar enthalpy of dissolution ( $\Delta_{sol}H$ , J mol-1) of amines instead of NH3.

Lines 196-200: "In this study, H of amines was also calculated based on Eq. (2) (Sander et al., 2015).

$$H(T) = H^{\Theta} \times \exp\left[\frac{-\Delta_{sol}H}{R}\left(\frac{1}{T} - \frac{1}{T^{\Theta}}\right)\right]$$
(2)

Where H(T) is the value of H at the specified temperature T (K),  $H^{\Theta}$  is H at standard temperature (298.15K),  $\Delta_{sol}H$  (J mol-1) is the molar enthalpy of dissolution, and R (8.314 J mol-1 K-1) is the gas constant."

(2) We have also modified the reference temperature in the original manuscript from 20°C to 25°C.

Lines 238-239: "

$$pK_a = pK_{wa} + (0.1552 - 0.0003142T)I$$
(9)

where,  $pK_{wa}$  is the value of  $pK_a$  in the pure water (T=25 °C, I=0 mol dm-3). "

Lines 801-802:"

| 5 1                                                          |                          |                          |                          |               |
|--------------------------------------------------------------|--------------------------|--------------------------|--------------------------|---------------|
|                                                              |                          |                          | — Pafarancas             |               |
| variable (Units)                                             | MMA                      | DMA                      | TMA                      | - References  |
| MW (g mol -1 )                                    | 31.06                    | 45.12                    | 59.11                    | Gibb et al.   |
|                                                              |                          |                          |                          | (1999)        |
| Henry's law constant, H (T=25 °C, I=0 mol dm -3 ) |                          |                          |                          | Sander        |
|                                                              | 0.00055                  | 0.00080                  | 0.0040                   | (2015); Leng  |
|                                                              |                          |                          |                          | et al. (2015) |
| Aqueous dissociation equilibria of atmospheric               |                          |                          |                          |               |
| amines (first second acid dissociation constants),           | 2.1878×10 -11 | 1.8621×10 -11 | 1.5849×10 -10 | Ge et al.     |
| $K_{w\alpha}$ (T=25 °C, I=0 mol dm -3 )           |                          |                          |                          | (2011b)       |
| $\Delta_r H_0 (kJ mol^{-1})$                                 | 53.737                   | 49.450                   | 36.017                   |               |
| $K_{\rm H} (M \text{ atm}^{-1}, T=25 \text{ °C})$            | 89.3*                    | 53.7*                    | 8.9*                     | C 1           |
| $d\ln H = \Lambda H$                                         |                          |                          |                          | (2015): Leng  |
| $\frac{d \Pi \Pi}{d(1/H)} = \frac{\Delta_{sol} \Pi}{R}$      | 4050*                    | 5200*                    | 5966*                    | (2013), Leng  |
| $u(1/11)$ $\Lambda$                                          |                          |                          |                          | et al. (2015) |

Table 4 Physicochemical parameters used in flux calculations.

\* Calculated from the data in references."

③ In addition, we have also modified the calculation of  $pK_a$ . We replaced  $pK_{wa}$ , which was a constant in Eq. (9) in the original manuscript, with a variable varying with temperature, and at the same time added a related expression in the revised manuscript (also refers to comment 3 by Reviewer 1).

Lines 239-244: "Moreover, Bell et al. (2008) pointed out that  $K_{wa}$  in Eq. (9) should not be a constant, but should depend on temperature. The equation for calculating  $K_{wa}$  is as follows (Ge et al., 2011b):

$$\ln(K_{wa}(T)) = \ln(K_{wa}(T_{r})) - \Delta_{r} H^{0} \left(\frac{1}{T} - \frac{1}{T_{r}}\right) / R$$
(10)

Where  $K_{wa}(T)$  is the value of the equilibrium constant at the specified temperature T(K),  $T_r$  is the reference temperature of 298.15 K,  $\Delta_r H^o$  (kJ mol-1) is the enthalpy change for the reaction at the reference temperature."

2) The authors use observed values of MMA, DMA and TMA dissolved in sea water from other sea areas and state that "all sites are located in densely populated areas". However, values from Hawaii or the Arabian Sea are used which are obviously not as densely populated areas as the Chinese coastline. Why do the authors use only these values, but neglect other measured values from Yang et al. (1994), Gibb et al. (1999), and van Pinxteren et al. (2019)?

**Response:**

We thank the reviewer for the constructive comments. We have added other measured values from recent publications to Table 3. In addition, we have also explained specific usage of the data in the table.

Lines 213-218: "In this study, the mean value of all  $[C_{(s) tot}^+]$  observations in the same quarter is used for the relevant calculation for the target month. If the observation time is not specified, the data is included in the calculation of the mean value; if the observation value is not specified (i.e., if only the variation range of observation values is given), the data is not included in the calculation. In the simulation period (July 2015, and December 2019), the  $[MMAH^+_{(s)tot}]$  was 36.1 nM and 38.9 nM, respectively,  $[DMAH^+_{(s)tot}]$  was 6.0 nM and 9.8 nM, and  $[TMAH^+_{(s)tot}]$  was 6.8 nM and 7.6nM, respectively."

**Lines 799-800: "**

| Table 3 A br | rief summary | of previou | s research | results or | n the distri | bution of | f amines. |
|--------------|--------------|------------|------------|------------|--------------|-----------|-----------|
|              | 2            | 1          |            |            |              |           |           |

| Ocean    | Location                          | Туре     | Period              | [NH 3 (g)]
(pmol m -3 ) | [MMA(
g)]
(pmol | [DMA(
g)]
(pmol | [TMA(g
)] (pmol
m -3 ) | [NH 3(s)
tot + ]
(nM) | [MMA (s
) tot + ]
(nM) | $[DMA_{(s}) + 1]$ | [TMA (s)
tot + ]
(nM) | Ref.                      |
|----------|-----------------------------------|----------|---------------------|--------------------------------------------------|-----------------------|-----------------------|-----------------------------------------|---------------------------------------------------|----------------------------------------------------|-------------------|---------------------------------------------------|---------------------------|
|          |                                   |          |                     | (                                                | m -3 )     | m -3 )     |                                         |                                                   |                                                    |                   |                                                   |                           |
| Pacific  | Hawaii                            | coastal  | July–August
1985 | -                                                | 11±5                  | 93±51                 | 30±19                                   | -                                                 | 52±20                                              | 1.5±20            | 12±3                                              |                           |
| Atlantic | Narragans
ett, Rhode
Island | coastal  | July–August
1985 | -                                                | 52±12                 | 240±40                | 100±40                                  | -                                                 | -                                                  | -                 | -                                                 | Van                       |
| Atlantic | Massachu
settes                | coastal  | November
1984    | -                                                | -                     | -                     | -                                       | -                                                 | 200±58                                             | 8.9±4.4           | 41±27                                             | Neste et
al.
(1987) |
| Atlantic | Massachu
settes                | coastal  | November
1985    | -                                                | -                     | -                     | -                                       | -                                                 | 32±5                                               | 8.9±1.1           | 10±13                                             | (1967)                    |
| Atlantic | Salt Pond                         | coastal  | November
1985    | -                                                | -                     | -                     | -                                       | -                                                 | 55±20                                              | 6.7±2.2           | 10±4                                              |                           |
| Atlantic | Flax
Pond,
New York         | seawater | -                   | -                                                |                       |                       |                                         |                                                   | 5-60                                               | 15-180            | <3-80                                             | Yang et
al.,1993       |
| Atlantic | Flax
Pond,
New York         | seawater | -                   | -                                                |                       |                       |                                         |                                                   | 5-40                                               | 25-180            | 10-50                                             | Yang et
al.,1994       |
| Atlantic | Mediterra
nean                 | offshore | -                   | -                                                |                       |                       |                                         | 33±9.6                                            | 7.5±5.5                                            | 4.6±3.0           | 1.4±1.6                                           | Gibb et                   |
| Atlantic | Mediterra
nean                 | coastal  | -                   | -                                                |                       |                       |                                         | 252±5
06                                       | 18±10.0                                            | 12±11.4           | 10±6.9                                            | aı.,
1994              |

| Atlantic | Plymouth
Sound,
UK | coastal  |                                                          | -                   |                 |                  |                    | 230-59
4       | 4-23                                 | 13-22                                | 4-17                                  |                 |
|----------|--------------------------|----------|----------------------------------------------------------|---------------------|-----------------|------------------|--------------------|-------------------|--------------------------------------|--------------------------------------|---------------------------------------|-----------------|
| Atlantic | Sutton
Harbour        | coastal  | -                                                        | -                   |                 |                  |                    | 7700              | 91                                   | <                                    | 15                                    | Gibb et
al., |
| Atlantic | Mediterra
nean        | offshore | -                                                        | -                   |                 |                  |                    | 22-60             | <-9                                  | <-9                                  | <-7                                   | 1995a           |
| Atlantic | Mediterra
nean        | coastal  | -                                                        | -                   |                 |                  |                    | 26-660            | 4-38                                 | 3-15                                 | 4-22                                  |                 |
| Atlantic | Arabian
Sea           | coastal  | 27 August–4
October
1994 (AS
series)            | (346-1726)94
6   | 37-177
(114) | 16-65*
(39)   | <1od -
1.4(0.9) | 80-150
(108)   | <30d-6
58
(16.2)               | <lod-13
.9
(4.54)</lod-13
 | <lod-0.
44
(0.044)</lod-0.
 |                 |
| Atlantic |                          |          | 16
November
-19
December
1994 (AS
series) | (2454-5628)3
780 | 50-241
(143) | 50-870*
(196) | <10d-1.
4 (8.1) | 81-253
(156)   | 10.1-49
.8
(25.6)              | <lod-9.
95
(4.85)</lod-9.
 | <lod-0.
8 (0.14)</lod-0.
       | Gibb et         |
| Atlantic | Arabian
Sea           | offshore | 27 August–4
October
1994 (A
series)             |                     |                 |                  |                    | 70-150
(115)   | <lod-13
.9
(5.24)</lod-13
 | <lod-11
.1
(3.85)</lod-11
 | <lod
(n/a)</lod
                | al.,
1999a   |
| Atlantic |                          |          | 10
November
-19
December
1994 (A             |                     |                 |                  |                    | 137-23
0 (230) | <lod-16
.5 (8.8)</lod-16
      | <lod-5.
7 (2.04)</lod-5.
      | <lod-0.
3
(0.036)</lod-0.
  |                 |
| Atlantic | Arabian
Sea           | coastal  | August–Oct
ober 1994
(AS series)                   |                     |                 |                  |                    | 139±1
35       | 12±20                                | 3.0±4.1                              | 0.10±0.
37                         |                 |
| Atlantic |                          |          | November
-December
1994 (AS
series)             |                     |                 |                  |                    | 206±2
71       | 22±13                                | 4.2±2.8                              | 0.45±0.
81                         | Gibb et         |
| Atlantic | Arabian
Sea           | offshore | August–Oct
ober 1994
(A series)                    |                     |                 |                  |                    | 91±91             | 6±7                                  | 2.9±2.8                              | 0.05±0.
21                         | al.,
1999b   |
| Atlantic |                          |          | November
-December
1994 (A
seires)              |                     |                 |                  |                    | 112±7
6        | 12±7                                 | 2.9±1.6                              | 0.13±0.
24                         |                 |

| Atlantic | Arabian
Sea                                                                                      | coastal  | August–Oct
ober 1994
(GOM
series) |  |
113±1
02 | 11±9          | 2.8±3.1       | 0.19±0.
42 | Gibb                                    |
|----------|-----------------------------------------------------------------------------------------------------|----------|--------------------------------------------|--|-----------------|---------------|---------------|---------------|-----------------------------------------|
| Atlantic | Ryder
Bay                                                                                        | seawater | February –
1999                         |  |                 | 12.0±
9.1  | 3.8±3.9       | 1.6 ±         | and
Hatton,
2004                  |
| Atlantic | Western
English
Channel                                                                       | seawater | January –
February
1999              |  |                 | 3             | 6             | 20            | Cree et
al.,
2018                 |
| Atlantic | the island
of Sao
Vicente                                                                     | seawater | November
2011
November
2013       |  |                 | 5-33
11-23 | 2-15
7-197 | -             | van
Pinxter
en et
al.,
2019 |
| Atlantic | the
regions of
Antarctic
Peninsula,
South
Orkney,
and South
Georgia
Islands | seawater | January-Feb
ruary, 2015                 |  |                 |               |               | 4.2           | Dall'Os
to et al.,
2019           |
|          |                                                                                                     |          |                                            |  |                 |               |               | 8.3
3.7    |                                         |
| Pacific  | the coastal
zones of
Dalian                                                                   | seawater |                                            |  |                 |               |               | 5.1
6.3    | Wu et
al.,
20s20                  |
|          |                                                                                                     |          |                                            |  |                 |               |               | 3.1           |                                         |
|          |                                                                                                     |          |                                            |  |                 |               |               | 3.2           |                                         |

••

3) Formula 10 In the publication of Khoo et al. (1977) the formula is different. It is pKa = pKwa + (0.1552 - 0.0003142T)\*I instead of the applied pKa = pKwa + (0.1552 - .003142T)\*I Furthermore, there was a correction of this prediction method by Bell et al. (2007; 2008). This has to be checked and the simulations have to be reperformed.

**Response:**

Thanks to the reviewer for the careful scrutiny. We have corrected the mistake in this equation. In addition, according to the modified method in Bell et al. (2008), the  $pK_{wa}$  calculation in this equation is supplemented. According to Bell et al. (2008)'s verification of Khoo et al. (1977)'s prediction method,  $pK_{wa}$  is not a constant at a certain reference temperature, but should vary with temperature. We adopt the method for calculation of temperature dependence of  $pK_{wa}$  in Ge et al. (2011b). Corresponding instructions have been added to the manuscript.

**Lines 236-244: "**

Further,  $K_a$  is also dependent on the temperature and ionic strength. For the NH3(s)–NH4+(s) system, the calculation of  $pK_a$  can be obtained from the empirical equation provided by Khoo et al. (1977) and applicable when the water salinity is less than 45.

$$pK_a = pK_{wa} + (0.1552 - 0.0003142T)I$$
(9)

Where  $pK_{wa}$  is the value of  $pK_a$  in the pure water (T=25 °C, I=0 mol dm-3). Moreover, Bell et al. (2008) pointed out that  $K_{wa}$  in Eq. (9) should not be a constant, but should depend on temperature. The equation for calculating  $K_{wa}$  is as follows (Ge et al., 2011b):

$$\ln(K_{wa}(T)) = \ln(K_{wa}(T_r)) - \Delta_r H^0 \left(\frac{1}{T} - \frac{1}{T_r}\right) / R$$
(10)

Where  $K_{wa}(T)$  is the value of the equilibrium constant at the specified temperature T(K),  $T_r$  is the reference temperature of 298.15 K,  $\Delta_r H^o$  (kJ mol-1) is the enthalpy change for the reaction at the reference temperature."

In addition, satellite-derived  $NH_3$  column concentration was added in this study. Combined with the algorithm modification of comments (1) and (2), and comparing with the calculation results in the original manuscript, the final MBE fluxes showed that the flux of amines exchange from the ocean to the atmosphere slightly decreased, while the flux from the atmosphere to the ocean slightly increased. This observation, however, does not significantly change the final conclusion.

Lines 297-310:

**"2.5.4 NH3**

Ammonia total columns retrieved from IASI measurements from the ANNI-NH3-v2.1R-I retrieval algorithm (https://iasi.aeris-data.fr/nh3/) was used to calculate gaseous amines concentration (e.g., MMA(g), DMA(g), TMA(g)). The empirical formula established by Yu et al. (2019) was used to estimate the ground NH3 concentration, as shown below.

$$[NH_3]_G = 0.3413 \times 10^{-15} \times [NH_3]_R$$
(12)

 $[NH_3]_G$  represents ground concentration measurements for NH3 (Fig. 2(d, i));  $[NH_3]_R$  represents NH3 column data (units: molec cm-2). In this study, the data of July 2015 and December 2016 (December 2019 data is incomplete) were adopted.

The estimates of MMA(g), DMA(g) and TMA(g) are based on the linear relationship between amine and ammonia established by Zheng et al. (2015) after field observations in the northern part of Nanjing from August to September 2012. The regression equation is shown below,

$$MMA_{(g)} = 0.85 \times NH_{3(g)} + 0.83 \tag{13}$$

$$DMA_{(g)} = 1.56 \times NH_{3(g)} + 1.28$$
 (14)

$$TMA_{(g)} = 0.37 \times NH_{3(g)} + 0.41$$
 (15)

where the unit of amines concentration is pptv, and the unit of NH3 concentration is ppbv."

**Lines 803-804: "**

Table 5 MBE fluxes of the three types of amines over ocean in July 2015 and December 2019.

| Site              | Date      | MMA (pmol m -2 s -1 ) | DMA (pmol m -2 s -1 ) | TMA (pmol m -2 s -1 ) | Ref.       |
|-------------------|-----------|---------------------------------------------|---------------------------------------------|---------------------------------------------|------------|
| Waters east of    | July 2015 | -0.81±0.90                                  | -1.9±1.7                                    | 2.8±1.3                                     | This study |
| China             | December  | 0 13+0 20                                   | 0 86±0 38                                   | 5 2+1 1                                     |            |
|                   | 2019      | -0.13±0.20                                  | -0.80±0.38                                  | 3.2 ±1.1                             |            |
| Coastal Hawaii    | -         |                                             |                                             |                                             | Van Neste  |
| and Massachusetts |           | 0.11-1.80                                   | - 0.460.49                                  | 0.20-3.20                                   | et al.     |
|                   |           |                                             |                                             |                                             | (1987)     |
| The island of Sao | November. |                                             |                                             |                                             | Van        |
| Vicente           | 2013      | 0.40-0.087                                  | 2.17-1.0                                    |                                             | Pinxteren  |
|                   |           | -0.40-0.087                                 | -2.17-1.9                                   | -                                           | et al.     |
|                   |           |                                             |                                             |                                             | (2019)     |
| "                 |           |                                             |                                             |                                             |            |

Lines 812-813: "

Figure 3: Spatial and temporal distribution of methylamines emissions: (a, d) MMA; (b, e) DMA; (c, f) TMA."

4) The authors state that chl-a influences the emission of amines into the atmosphere, but concentrate only on the pH effect. In environments rich of biological activity such as the sea-surface microlayer DMA concentrations can be up to one order of magnitude larger than in the bulk (van Pinxteren et al, 2019). A sensitivity study dealing the

**possible effect of higher chl-a on dissolved amine concentrations is missing and has to be done.**

**Response:**

Thanks for the constructive comments. The biochemical activity in the sea-surface microlayer does have a significant effect on the dissolution of amines in seawater. However, many of the biochemical processes involved in these processes are complex, and hence, there is still no effective algorithm for quantitative analysis of these processes. Therefore, only the intermediate variable pH in the empirical equation (Eq. (7)) involving Chla in this study was analyzed to assess the effect of Chla on amine dissolution. We have also added a sensitivity study about the synergistic effects of Chla, SST, and SSS on pH, pKa, and the concentration of unionized amine molecules.

**Lines 345-372: "**

**(b) Chla**

With the increase of Chla, the direction of amines exchanges between the ocean and atmosphere showed a trend of transferring from the atmosphere to the ocean (Fig. 5 (b, i)). According to Eq. (7), the increase of Chla will lead to the decrease of pH (Fig.6(a, e)), and hence, it is not conducive to the emission of amines. Chla is used to indicate primary production in the water body, and high Chla indicates a significant increase in phytoplankton. Water eutrophication is caused by the massive growth of phytoplankton due to the continuous importation of anthropogenic nutrients into the sea by rivers. Thus, the increased organic matter is transported to the subsurface water by settling and being decomposed by microorganisms. This process consumes oxygen in the water and forms a hypoxic environment. With the mixing of the water, the pH of the water changes. Zhao et al. (2020), based on the observation data of summer voyage from the Pearl River Estuary to the northern continental shelf of the South China Sea, found that the water on the west side of the Pearl River Estuary with obvious mixing of fresh water and seawater is characterized by low dissolved inorganic carbon (DIC) and high pH, while for the area with 20-30 meters water depth outside the mixing area, it is characterized by high DIC and low pH. The main source of amines comes from the degradation of organic matter in sediments (Carpenter et al., 2012), and therefore, an increase in Chla might mean more acidification in the ocean, making amines more soluble in seawater.

(c) SSS

It can be seen from Fig. 5 (c, j) that the increase of SSS will increase the tendency of the amines to be emitted from the water surface. It can be inferred from Eq. (8), (9), (10), and (11) that the increase of SSS will further inhibit the ionization of amines in seawater (Fig. 6((b, f)), which makes it more prone to emission from the water surface, resulting in an increase of the exchange fluxes.

**(d) SST**

An increase in SST will lead to a decrease of the fluxes of amines emitted from the ocean to the atmosphere (Fig. 5 (d, k)). As can be seen from Eq. (7), the increase of SST will lead to the decrease of pH (Fig.6(a, e)) and the reduction of amine pKa (Fig.6(b, f)), which makes the amines more likely to dissolve in water (Fig.6(c, g)). Due to the high stability of the marine

environment, the variation range of SST itself is small; therefore, SST has the least influence on the exchange flux among the four elements. It should be noted that El Niño conditions occurred by late May in 2015, which increased the global average temperature and affected the weather patterns in the study area (Kennedy et al., 2016). The annual average SST of our study areas was 0.5-1.0°C higher than the average value recorded during the period from 1961 to 1990 (Kennedy et al., 2016). Figure 5(d ,k) shows that the exchange fluxes of amines are negatively correlated with SST, but the sensitivity is low."

Lines 817-820: "

---

## Author Comment (AC2)

**Manuscript id: acp-2022-394**

**Manuscript title:** Contribution of marine biological emissions to gaseous methylamines in the atmosphere: an emission inventory based on satellite data

**Responses to Reviewers' comments:**

We thank the Reviewer for the constructive suggestions and helpful comments. We provide below itemized responses to each of the Reviewer's comments. The comments are given in bold while responses are in normal font. Changes made to the manuscript are shown in blue.

**Reviewer #2:**

**Overall impression:**

The study aims to estimate the gas phase concentration of three different amines using WRF-Chem. In addition to previous similar attempts using anthropogenic amine emissions only, the authors apply an online marine biogenic amine emission scheme. This new method seems sound and valid. However, the manuscript lacks of a thorough validation. (Some) Measurement results of different studies are presented in a table but never discussed in context with the simulated concentrations. Furthermore, the conclusions and discussion is not clearly formulated. For example, it is often unclear what type of aggregation / averages are referred to when presenting relative differences. This made it hard to follow the discussion and the results and likely lead to misunderstandings. Finally, potential shortcomings of the current method (e.g. loss to atmospheric aerosols is not considered) is not discussed anywhere in the paper.

Despite the fact that the manuscript could be improved in terms of language and clearer formulations at many points, the manuscript also needs major revision in the interpretation of results, discussion and conclusion. Since measured concentrations were presented, an evaluation section that compares these measurements, in particular the ones of the same year, with the modelled concentrations.

1) l. 138-139: Why is loss to aerosols not considered? Since it was correctly described in the introduction that amines play a role in aerosol formation, there should be a considerable loss to aerosols. This should be discussed here and in the conclusions. Response:

We agree with the reviewer's opinion that absorption by aerosols is an important sink of amines. However, in this study, we used the aerosol scheme of CAM\_MAM3\_AQ which is unable to include the loss of amines to aerosols at present. There are some studies (e.g., Yu and Luo, 2014 and Mao et al., 2018) have considered the aerosol adsorption of amine simply by changing the uptake coefficient (γ) of amines to aerosol (instead of considering new particle formation involving amines) using different aerosol schemes (WRF-Chem with CB05 scheme by Mao et al. (2018) and GEOS-Chem v8.3.2 with an advanced particle microphysics (APM) model by Yu and Luo(2014)). However, with the CAM\_MAM3\_AQ employed in this study, more complex parameters are needed to consider the loss of amines to aerosol instead of simply including uptake coefficient. In this study, we used the aerosol scheme of CAM\_MAM3\_AQ because it is possible to new particle formation with amines (although not done yet in this study).

At present, only three species, namely  $H_2SO_4$ ,  $NH_3$  and MSA, are taken into consideration in CAM\_MAM3\_AQ. The uptake rate (uptkrate) of the latter two by aerosol is obtained by multiplying the uptake rate of  $H_2SO_4$  into aerosol by a fixed value (uptkrate(NH\_3)= uptkrate (H\_2SO\_4)×2.08, uptkrate(MSA)=uptake(H\_2SO\_4)×1.28). The uptake rate of  $H_2SO_4$  is related to the particle size range, the concentration distribution of different particle sizes, and the diffusivity of  $H_2SO_4$  gas molecules. At present, there is not enough literature to obtain accurate parameters related to different amines and  $H_2SO_4$  in particles with different particle sizes. We plan to establish a new particle nucleation mechanism involving amines and embed it in the model based on laboratory simulation in a subsequent study. However, the current study did not consider the influence of aerosol absorption on amines. We have added relevant statements in the manuscript to explain why we did not consider aerosol absorption.

Lines 164-170: "Although absorption by aerosols is an important sink of amines, the current study did not consider the influence of aerosol absorption on amines. At present, only three species, namely  $H_2SO_4$ ,  $NH_3$  and MSA, are taken into consideration in CAM\_MAM3\_AQ. The uptake rate (uptkrate) of the latter two by aerosol is obtained by multiplying the uptake rate of  $H_2SO_4$  into aerosol by a fixed value (uptkrate( $NH_3$ ) = uptkrate( $H_2SO_4$ )×2.08, uptkrate(MSA)=uptake( $H_2SO_4$ )×1.28). The uptake rate of  $H_2SO_4$  is related to the particle size range, the concentration distribution of different particle sizes, and the diffusivity of  $H_2SO_4$  gas molecules. At present, there is not enough literature to obtain accurate parameters related to different amines and  $H_2SO_4$  in particles with different particle sizes."

2) l. 145-146: It is stated that 2015 was chosen since it had ,more consecutive days with field observations', however, these field observations were not used in the manuscript for validation purposes. Furthermore, the authors cite e.g. Zheng et al., 2015, in this context, who performed measurements in 2012, so this study probably does not help to explain why 2015 was chosen.

**Response:**

Thanks for the comment. We are aware of the flaws in the setting of simulation periods in the manuscript. Therefore, we abandoned the discussion on seasonal variation and re-selected the simulation period for the following reasons:

(1) Although January, April, July and October were selected as representative months of winter, spring, summer and autumn for China (Cai et al., 2017) in the original manuscript, the actual simulation period was short and could not adequately represent the changing characteristics of a season.

(2) Insufficient observational data are available to evaluate the model's performance of the simulation of amines in different seasons. It is therefore doubtful whether the model's results in January, April and October are representative of the actual seasonal variation of atmospheric amines. Therefore, after the acquisition of new continuous observations, we believe that the simulation period should be re-selected based on the observations that can be used for verification.

③ In the original manuscript, the expression of seasonal variation of the amines simulation results is mainly related to the wind direction, and there are relatively detailed expressions only for July when the wind blows from the sea to the land, while the results of the other three simulation periods when the prevailing wind blows from the land to the sea are less expressed. In addition, the newly selected prevailing winds for December 2019 also blow from land to sea, so reselecting the simulation period does not affect the manuscript's main conclusions.

④ In addition, the new observational data are from marine sites, and hence, we can find out how the simulation improves for different types of sites after the addition of MBE.

The simulation period was reset according to the observation data obtained from the literature research. The continuous observation data of Shanghai urban site and Yellow and Bohai Seas marine site were selected for the verification of the simulation results. Thus, the effect of the amine emission inventory established in this study on the simulation effect of different types of sites was quantified. Because we re-selected the simulation period in the process of revising the manuscript and abandoned the expression of seasonal changes in the original manuscript, the description of seasonal changes in the applied satellite data was deleted.

Lines 172-177: "In this study, continuous observation data from Yao et al. (2016) collected in Shanghai urban site on 25-31 July 2015 (MMA, DMA, TMA were observed), and Chen et al. (2021) collected in the Yellow and Bohai Seas during 9–22 December 2019 (DMA, TMA were observed) were used for verification. Therefore, we simulated the amines concentration in July 2015 (2015.7.22 00:00:00 to 2015.7.31 18:00:00), and December 2019 (2019.12.6 00:00-2019.12.22 18:00) to explore the changes at urban and marine sites."

Lines 826-828: "